# Structural and dynamic mechanisms of GABA$_A$ receptor modulators with opposing activities

Shaotong Zhu[1,4], Akshay Sridhar[2], Jinfeng Teng[1], Rebecca J. Howard [3], Erik Lindahl [2,3] & Ryan E. Hibbs [1] ✉

γ-Aminobutyric acid type A (GABA$_A$) receptors are pentameric ligand-gated ion channels abundant in the central nervous system and are prolific drug targets for treating anxiety, sleep disorders and epilepsy. Diverse small molecules exert a spectrum of effects on γ-aminobutyric acid type A (GABA$_A$) receptors by acting at the classical benzodiazepine site. They can potentiate the response to GABA, attenuate channel activity, or counteract modulation by other ligands. Structural mechanisms underlying the actions of these drugs are not fully understood. Here we present two high-resolution structures of GABA$_A$ receptors in complex with zolpidem, a positive allosteric modulator and heavily prescribed hypnotic, and DMCM, a negative allosteric modulator with convulsant and anxiogenic properties. These two drugs share the extracellular benzodiazepine site at the α/γ subunit interface and two transmembrane sites at β/α interfaces. Structural analyses reveal a basis for the subtype selectivity of zolpidem that underlies its clinical success. Molecular dynamics simulations provide insight into how DMCM switches from a negative to a positive modulator as a function of binding site occupancy. Together, these findings expand our understanding of how GABA$_A$ receptor allosteric modulators acting through a common site can have diverging activities.

GABA$_A$ receptors, the ionotropic targets of the major inhibitory neurotransmitter GABA, are pentameric ligand-gated ion channels that mainly function to suppress excitability in the central nervous system. Upon GABA binding, the intrinsic ion channel opens, which in most cases enables the influx of chloride to oppose depolarization and inhibit neuronal firing. Several neurological and neurodevelopmental disorders, including epilepsy, schizophrenia, autism, and anxiety disorders, have been associated with GABA$_A$-receptor dysregulation[1–4]. The diverse physiological roles of this receptor are related to variable expression of GABA$_A$-receptor subunits[5], with the most abundant synaptic isoform in adult brain composed of two α1-subunits, two β-subunits and one γ2-subunit[6].

The GABA$_A$ receptor is the principal site of action of benzodiazepines. These drugs act through two distinct groups of binding sites: a primary high-affinity site at the α/γ extracellular interface and three low-affinity binding sites located between subunits (γ/β and β/α) in the transmembrane domain[7–10]. Classical benzodiazepines like diazepam are used to treat epilepsy, insomnia, and anxiety disorders. However, administration of these drugs is often accompanied by the development of side effects including sedation, cognitive impairment, addiction, and tolerance, which is in part due to their non-selective modulation of many GABA$_A$-receptor subtypes[11]. Zolpidem, a representative of a new generation of benzodiazepine-site modulators, is the most widely prescribed

[1]Departments of Neuroscience and Biophysics, University of Texas Southwestern Medical Center, Dallas, TX 75390, USA. [2]Dept. of Applied Physics, Science for Life Laboratory, KTH Royal Institute of Technology, Solna, Sweden. [3]Dept. of Biochemistry and Biophysics, Science for Life Laboratory, Stockholm University, Solna, Sweden. [4]Present address: Institute for Protein Innovation, 4 Blackfan Circle, Boston, MA 02115, USA. ✉e-mail: ryan.hibbs@utsouthwestern.edu

hypnotic in the United States. It is an imidazopyridine, chemically distinct from benzodiazepines, but exerts similar pharmacological and physiological effects through GABA$_A$ receptors. Interestingly, zolpidem shows significantly higher binding affinity and modulatory potency at α1β2 receptors, exhibiting high selectivity for α1-containing GABA$_A$ receptors[12]. However, direct structural information on zolpidem binding to GABA$_A$ receptors that would illuminate how this selectivity is determined is currently absent.

Functionally diverse ligands interact with the benzodiazepine site in the extracellular α/γ-subunit interface of the GABA$_A$ receptor (Fig. 1a)[13,14]. Classical benzodiazepines and zolpidem are positive allosteric modulators (PAM) that enhance the response to GABA. Flumazenil can have variable activity but is mainly considered a silent allosteric modulator, with limited direct effects on GABA-induced currents, and antagonizes the action of benzodiazepine PAMs. Several β-carbolines such as methyl-6,7-dimethoxy-4-ethyl-β-carboline-3-carboxylate (DMCM) are negative allosteric modulators (NAM) that can decrease GABA$_A$-receptor activity, particularly at low concentrations. Conversely, at high concentrations DMCM can act as a positive modulator, especially in the presence of flumazenil, to enhance GABA-elicited currents[15–19]. Identification of structural elements that couple the benzodiazepine-site binding to modulation of channel function has begun[20–23]. However, the mechanisms by which DMCM binding is transduced to both potentiation and inhibition remain unclear.

Here we obtained cryo-EM structures of the α1β2γ2 GABA$_A$ receptor in complex with GABA plus either zolpidem or DMCM, enabling visualization of the binding sites and protein interactions of these important ligands (Fig. 1b–e). A combination of structural pharmacology and metadynamics simulations supports a rationale for zolpidem selectivity for the α1β2 subtype of GABA$_A$ receptors. Furthermore, structural comparisons and all-atom simulations suggest a mechanism for the distinctive bimodal modulation profile of DMCM. This structural and simulation data shed new light on allosteric ligand binding and modulation in the GABA$_A$-receptor family, with applications in channel biophysics and the refinement of benzodiazepine-site therapeutics.

## Results and discussion
### Zolpidem recognition
Z-drugs including zolpidem, zaleplon, zopiclone and eszopiclone are members of a new generation of sedative-hypnotic drugs that are benzodiazepine-like GABA$_A$ receptor modulators. These drugs promote sleep by acting through the same benzodiazepine sites to potentiate GABA$_A$ receptors. However, they are chemically distinct from benzodiazepines. Compared to classical benzodiazepines, Z-drugs have less impact on sleep architecture, and induce a pattern and quality of sleep similar to natural sleep[24]. Both benzodiazepines and Z-drugs cause adverse effects; however, benzodiazepines carry greater risk of tolerance and abuse. In clinical practice, prescriptions

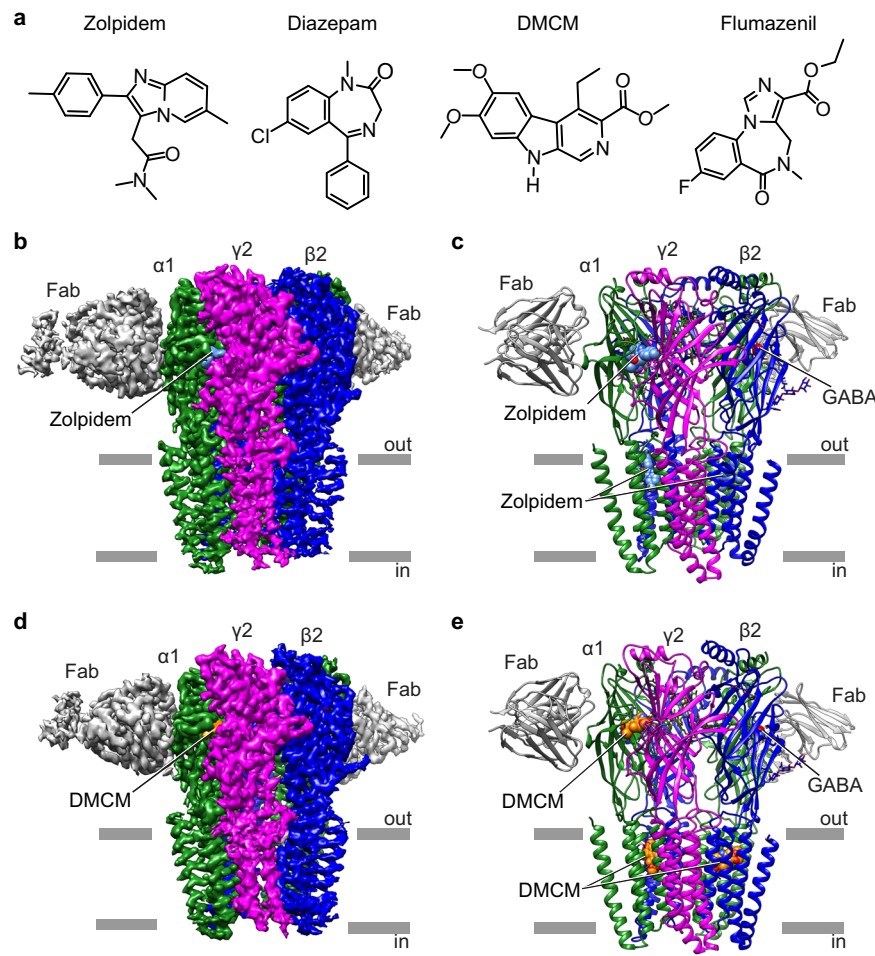

**Fig. 1 | Representative benzodiazepine-site ligands and overall structures of the GABA$_A$ receptor-ligand complexes. a** Zolpidem is an imidazopyridine; diazepam and flumazenil have a benzodiazepine core; DMCM is a β-carboline. **b**, **c** Side views of the 3D reconstruction and atomic model of GABA$_A$ receptor-zolpidem complex colored by subunit: α1-green, β2-blue, γ2- magenta and Fab-gray. **d**, **e** Side views of the 3D reconstruction and atomic model of GABA$_A$ receptor-DMCM complex colored as in **b**. Zolpidem is shown as cyan spheres, DMCM as orange spheres, GABA as red spheres.

for Z-drugs have grown rapidly in the past decades, surpassing benzodiazepines for the treatment of insomnia[25]. The structural elements and modulation mechanisms underlying Z-drug action are not understood and are of great interest due to their clinical importance.

To investigate the structural basis for subtype-specific binding by Z-drugs, we first purified a modified α1β2γ2 GABA$_A$-receptor construct bound with zolpidem plus GABA and reconstituted the receptor-ligand complex into lipid nanodiscs (Methods)[20]. Fab fragments were used to facilitate particle alignment. The modified receptor construct, lacking the intracellular domain to improve expression level and biochemical behavior, corresponds with published observations for both GABA activation and zolpidem potentiation (Fig. 2a)[26,27]. Flumazenil, a neutral α1β2γ2 benzodiazepine antagonist, blocked zolpidem's positive modulation of the receptor (Fig. 2a, b). Initial EM data processing resulted in a 2.7 Å resolution map with weak density in the γ2 subunit TMD, consistent with our previous structural studies on the synaptic GABA$_A$ receptor[20,21] (Supplementary Fig. 1a–f and j). Focused classification on this region resulted in an overall 2.9 Å resolution map with improved resolution in the TMD, which enabled model building and refinement of the entire receptor (Fig. 1b, c and Supplementary Fig. 1g–i and k).

The density for zolpidem was particularly well defined at the extracellular α/γ benzodiazepine site, which allowed us to position the ligand with confidence (Fig. 2c–e, Supplementary Fig. 2e). Zolpidem has a T-shape, with its imidazopyridine-benzene ring axis roughly parallel to the α1-subunit Loop C, its pyridine methyl group pointing toward α1-H102, and its methylbenzene facing the complementary γ2 subunit (Fig. 2d, e). The dimethylacetamide arm buries deep into the subunit interface, with the carbonyl oxygen pointing out toward the tip of α1 Loop C. The imidazopyridine ring is sandwiched between two aromatic residues α1-Y210 and γ2-F77, likely forming π stacking interactions. Mutating these two residues to alanine and leucine respectively has been shown to decrease ligand affinity, while substituting F77 with tyrosine slightly improved it; thus, aromatic interactions with these two residues are important for stabilizing binding[28,29]. The γ2-Y58 residue packs against the zolpidem acetamide group. On the α1-subunit Loop C, S205 is positioned to form a hydrogen bond (2.7 Å) with the zolpidem carbonyl group (Fig. 2d, Supplementary Fig. 2d). Furthermore, T207 on the same loop forms another hydrogen bond with the imidazole nitrogen (Fig. 2d, Supplementary Fig. 2d). Mutating S205 to cysteine was previously shown to reduce zolpidem affinity 7-fold, while changing T207 dramatically

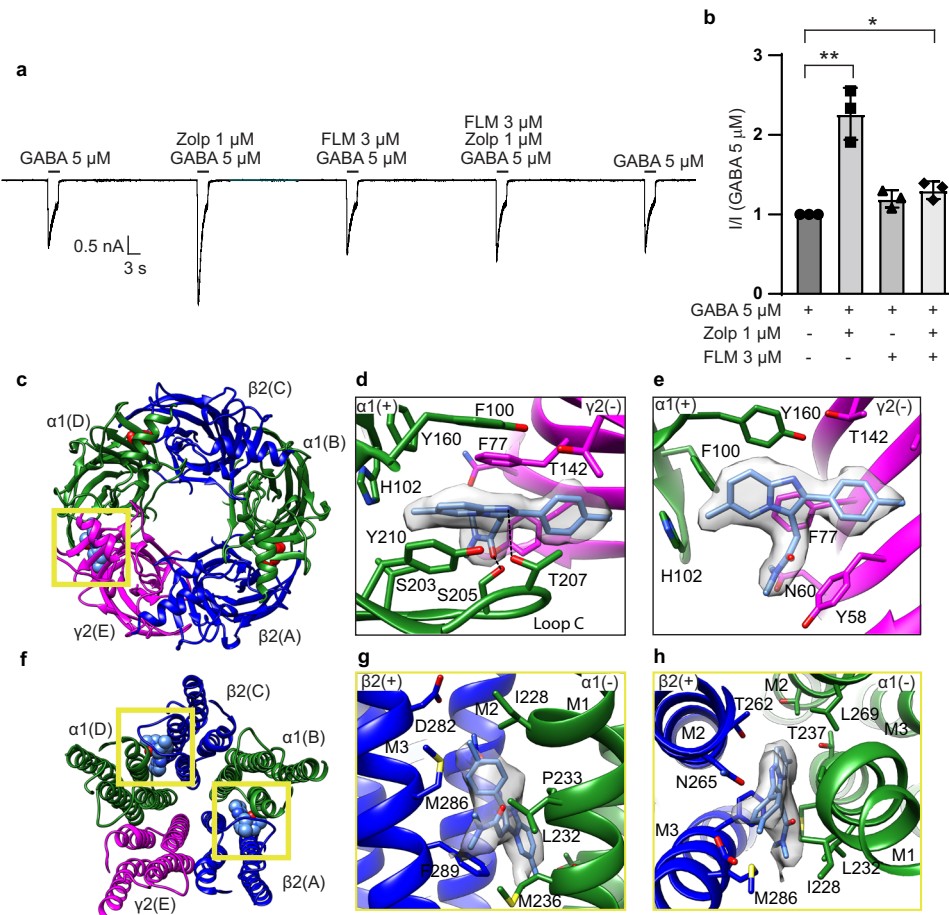

**Fig. 2 | Zolpidem activity and binding sites. a** Representative patch-clamp electrophysiology of the cryo-EM construct recorded from three independent cells, showing zolpidem potentiates the GABA response and flumazenil blocks this potentiation. **b** Statistical analysis of recordings as in **a** from three independent cells. Bar graph shows mean responses with standard deviation for the fractional zolpidem potentiation with and without flumazenil. Two-sided Welch's t-test was used. A p-value of ≤0.05 was considered statistically significant. *p = 0.042; **p = 0.022. **c** Atomic model of ECD viewed from synaptic side; yellow box indicates benzodiazepine binding site bound with zolpidem shown as spheres. **d, e** Detailed architecture of zolpidem binding pocket boxed in **c. d** Synaptic view as in **c. e** Side view of the zolpidem binding pocket with loop C backbone hidden for clarity. **f** Atomic model of TMD viewed down channel axis from synaptic side; yellow boxes highlight two approximately equivalent zolpidem binding sites with ligands shown as spheres. **g, h** Zolpidem binding site details at one TMD β/α interface. **g** Side view of the zolpidem binding pocket. **h** Synaptic view as in **f**. Principal side is "+" and complementary side is "−". H-bonds indicated with dashed line. Source data are provided as a source data file.

disrupted drug efficacy[27], supporting distinctive roles for these residues in mediating ligand activity.

Mutagenesis, electrophysiology, and structural studies have revealed multiple binding sites for benzodiazepine-type ligands on GABA$_A$ receptors[8,20,23]. In addition to the classical α/γ ECD site (Fig. 2c–e), we observed two strong densities at the β2/α1 TMD interfaces consistent with the T-shape of zolpidem (Fig. 2f–h, Supplementary Fig. 2h, i). The positions of these two approximately equivalent sites overlap with two of the three TMD sites previously observed for diazepam (Fig. 3d–f). In these pockets, the imidazopyridine-benzene ring axis of zolpidem is roughly parallel to the channel axis, with its methylbenzene group pointing extracellularly and its acetamide arm orienting away from the pore. Residues M286 and F289 from the β2 subunit straddle the dimethylacetamide group, forming van der Waals contacts with the ligand. Zolpidem is further stabilized by hydrophobic interactions between its imidazopyridine ring and P233 on the α1-subunit M1 helix (Supplementary Fig. 2g). Whereas TMD binding has been proposed to underlie biphasic modulatory properties of diazepam, its functional role in zolpidem modulation remains unclear[8]. Interestingly, whereas diazepam has a third TMD-binding site at the γ2/β2 interface, no density was evident for zolpidem at this location in our EM maps. This result suggests that the γ2/β2 TMD site is of little importance for zolpidem activity.

## Zolpidem selectivity

A distinguishing property of zolpidem is its selectivity among both α and γ GABA$_A$-receptor subunits. Further, in contrast to classical benzodiazepines, zolpidem can exert hypnotic effects at lower doses than its other neuropharmacological effects, including muscle relaxation and anti-convulsant activity[30]. This differential response may be attributed to zolpidem selectivity for GABA$_A$-receptor subtypes. Except for α4- and α6-containing GABA$_A$ receptors, classical benzodiazepines bind GABA$_A$ receptors containing all α-subunit isoforms to a similar degree[31,32]. Conversely, zolpidem displays a preference for α1-containing GABA$_A$ receptors, intermediate affinity for α2- and α3-containing receptors, and almost no affinity for α5-containing receptors[31–33]. The sedative component of benzodiazepines has been suggested from rodent studies to be mediated by α1-containing GABA$_A$ receptors, with other effects like anxiolysis mediated by other α subunit isoforms[34]. While clinical studies in humans reveal a more complex scenario[35], the α1-selectivity may still explain why zolpidem is clinically useful as a hypnotic. The γ subunits contribute to zolpidem selectivity as well, in which the ligand exhibits lower affinity for γ1- and γ3-containing subtypes compared to γ2-containing receptors[36–39].

Comparing zolpidem and diazepam-bound cryo-EM structures allowed mapping of structural determinants of zolpidem selectivity (Fig. 3). In its TMD sites, zolpidem occupies the same general pocket as diazepam, with contacts more distributed between the β2-M3 and α1-M1 helices (Fig. 3e, f). Moreover, zolpidem contacts in the TMD β/α interfaces are conserved among α subunits, indicating they are not likely to contribute to selectivity among subtypes (Fig. 3i). Therefore, for purposes of understanding selectivity, we focused on the canonical α/γ benzodiazepine site in the ECD. Superposition of the zolpidem complex with that previously reported for the α1β2γ2 receptor with diazepam showed that backbone atoms of the two complexes are well aligned in the ECD-benzodiazepine site, and that aromatic residues forming the binding pocket adopt similar orientations (Fig. 3b, c).

A clear ECD feature of our zolpidem complex is a close contact between the ligand's imidazopyridine methyl group and α1-H102 (Fig. 2d, e). Previous structural and functional studies found that interactions with α1-H102 are important for the activity of several benzodiazepine ligands (diazepam, alprazolam, and flumazenil)[20,21,23]. However, whereas α1-H102 directly contacts the chlorine atom of diazepam, the aligned moiety of zolpidem is a methyl group, with less

favorable electrostatic interactions (Supplementary Fig. 2d). Indeed, nearly all benzodiazepine-site ligands (diazepam, flumazenil, DMCM, zaleplon) except zolpidem contain a chemical group (such as Cl or F) that could accept a hydrogen bond from this histidine. Accordingly, the identity of α1-H102 is essential for classical benzodiazepine binding, but less critical for zolpidem sensitivity[40]. Thus, while α1-H102 contributes to binding zolpidem as well as other modulators, additional interactions are involved in the α1 selectivity for zolpidem[27]. In α4 and α6 subtypes, the equivalent residue is the notably larger, more basic arginine, which would be expected to clash with atoms of either zolpidem or diazepam (Fig. 3g). Another divergent residue, α1-T163, has also been proposed to zolpidem specificity[41]. However, it does not directly contact zolpidem in our structure, suggesting this residue's effect on ligand selectivity is more complex. To quantify the importance of these contacts, we performed molecular dynamics simulations of zolpidem binding at the benzodiazepine site, enhanced by funnel metadynamics to improve sampling of the free-energy landscape in the region surrounding the α/γ ECD interface (Supplementary Fig. 2a–c)[42]. Simulations of the wild-type receptor verified a free-energy well for zolpidem, centered at its position in the cryo-EM structure (Fig. 3j). In contrast, simulations with the mutation H102R resulted in a diffuse free-energy landscape, consistent with disrupted binding at the active site (Fig. 3l). In the context of the mutation T163P, the landscape was similar to wild type, supporting an indirect role for this modification on zolpidem modulation (Fig. 3k).

Residues on α1 Loop C are not well conserved (Fig. 3g); mutating a subset of these residues has been shown to impair zolpidem binding affinity and potentiation, which suggests this region plays a major role in defining zolpidem subtype selectivity[31,41,43]. Residue V203 is unique among α subunits and the equivalent position in α2-α6 subunits is an isoleucine (Fig. 3g). Substitution of this residue with cysteine has been shown to ablate zolpidem affinity and efficacy[44,45]. S205, also on Loop C, is oriented to form a hydrogen bond with the zolpidem carbonyl group. In the α5 subunit, the equivalent residue is a threonine; serine substitution in α5 dramatically increased the zolpidem affinity[41]. Besides these two residues, G201 is also important for zolpidem recognition. It is located on Loop C but is distant from the ligand binding site. In α2-α6 subunits, its equivalent residue is glutamate, which has a larger, charged sidechain. Mutating α1-G201 to glutamate impaired zolpidem affinity, while changing E201 in the α5 subunit to glycine increased sensitivity to zolpidem[46]. Thus, several residues on Loop C act in concert to decrease zolpidem affinity, particularly in α5-containing GABA$_A$ receptors.

The γ subunits also contribute to zolpidem selectivity, as the drug binds preferentially to γ2-containing receptors[36–39]. At the benzodiazepine site, zolpidem adopts a shallower binding pose compared to diazepam, but makes more contacts with the γ2 subunit (Fig. 3b, c). In particular, the phenyl ring of F77 on the γ2 subunit forms hydrophobic interactions with zolpidem's imidazopyridine ring. The γ1 subunit has an isoleucine residue in the position homologous to this residue (Fig. 3h). Exchange of the aromatic ring by branched hydrophobic side chains such as isoleucine has been shown to reduce zolpidem affinity >600-fold, compared to only a 5-fold loss for diazepam[28]. As for the H102R mutation, funnel metadynamics showed a diffuse free-energy surface (Fig. 3m) for zolpidem binding in the presence of F77I, indicative of destabilization compared to the structurally resolved configuration. Another non-conserved residue, γ2-M130 (homologous to γ1-L132 and γ3-L133), has been identified to be important for recognition of zolpidem, but not diazepam[37]. In our model, this residue is not in direct contact with the ligand. γ2-M130 is located at the top of the binding pocket with its side chain extending toward zolpidem; however, it is still too far away (>4 Å) to have a direct impact on ligand positioning. Based on our data, M130 may contribute to zolpidem selectivity indirectly by helping to form a local chemical environment that is selectively favorable for zolpidem binding.

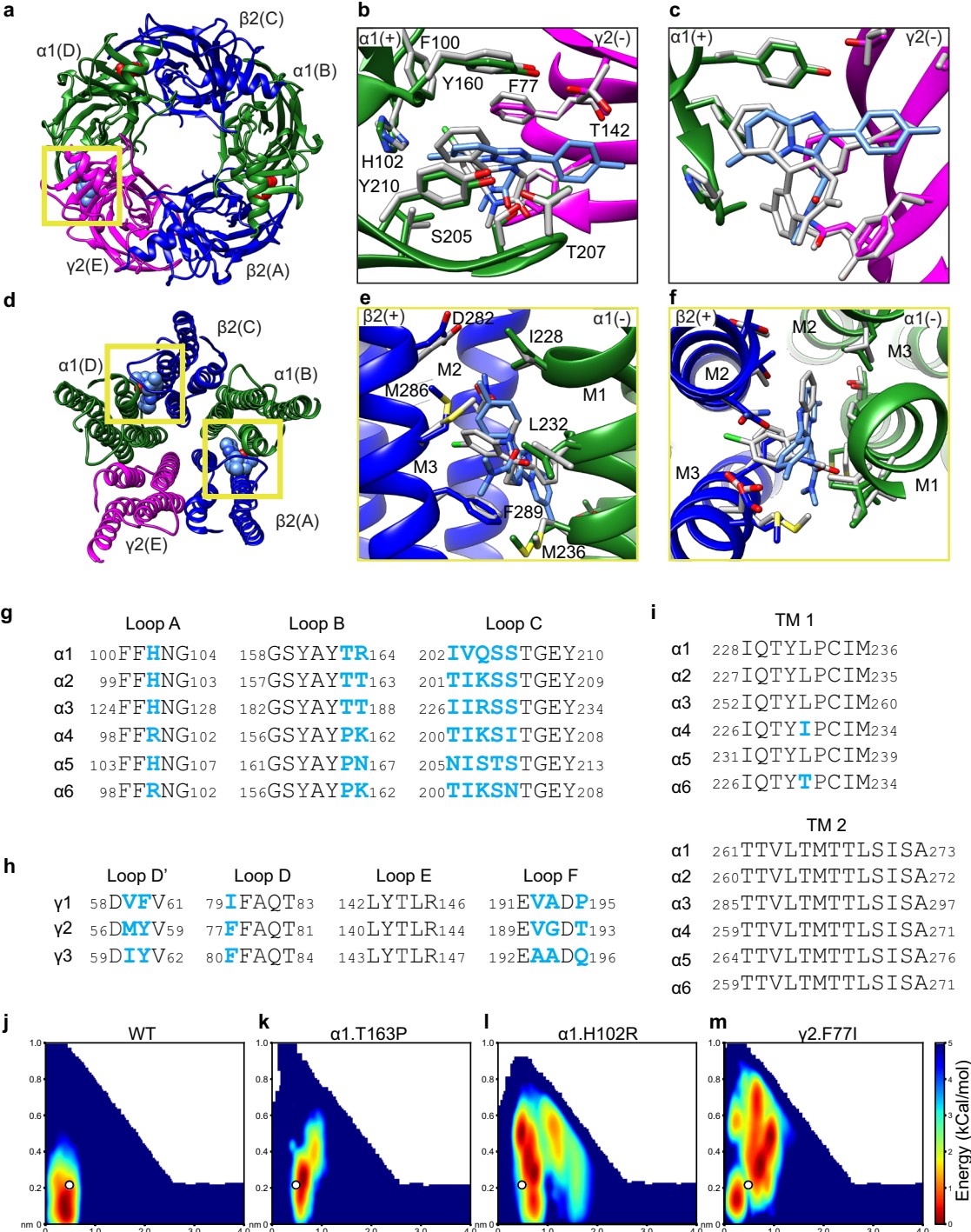

**Fig. 3 | Zolpidem versus diazepam binding. a** Synaptic view; yellow box indicates the ECD-benzodiazepine site. **b**, **c** Superposition of bound diazepam (in gray) on zolpidem-bound structure. **b** Synaptic view as in **a**. **c** Side view of the zolpidem binding pocket with loop C backbone hidden for clarity. **d** Atomic model of TMD viewed down channel axis from synaptic side; yellow boxes highlight two zolpidem binding sites with ligands shown as spheres. **e**, **f** Superposition of bound diazepam (in gray) on zolpidem-bound structure, focused on β/α interface. **e**, Side view as in Fig. 2g. **f** Synaptic view as Fig. 2h. **g**–**i** Sequence alignment of the ECD loops and TMD helices involved in ligand binding pockets. The canonical benzodiazepine site at α/γ interface is formed by residues located on several discontinuous segments termed as "Loops A–F"[78]. **g** Loops A–C in ECD. **h** Loops D–F in ECD. **i** TMD helices 1 and 2. Blue residues are non-conserved residues among α1 to α6 subunits and γ1-γ3 subunits. Diazepam-bound model, PDB 6X3X. **j**–**m** Two-dimensional free-energy profiles for zolpidem binding at ECD site calculated from funnel metadynamics. Axes are distances in nanometers. The ligand position corresponding to the receptor structure is illustrated as a white dot. **j** wildtype. **k** α1-T163P. **l** α1-H102R. **m** γ2-F77I. Source data are provided as a Source Data file.

In conclusion, zolpidem selectivity leverages its distinct chemical structure and interactions with surrounding residues. In the absence of a strong interaction with α1-H102, the ligand depends more on other elements of the local chemical environment for its high-affinity binding. The poor sequence conservation in loop C of the α subunits begets distinct, subunit-dependent local conformations at the binding site, which contributes to zolpidem's preference for α1-containing receptors. In addition, the γ2 subunit

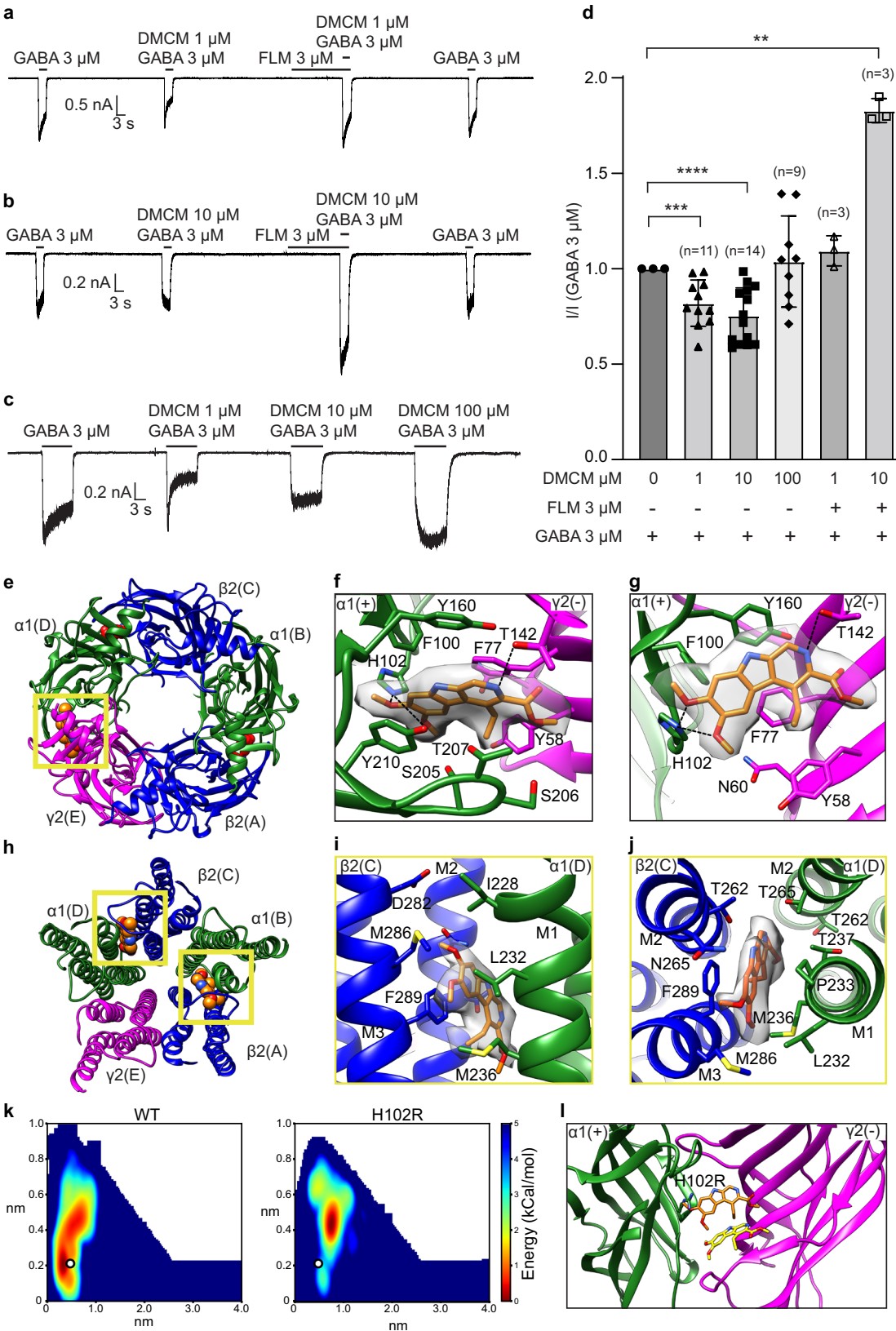

makes extensive contact with zolpidem, helping to fine tune binding specificity.

## DMCM recognition
DMCM has potent convulsant and anxiogenic properties in rodents. It modulates GABA_A receptors in a biphasic manner. The drug at low concentrations acts as a negative allosteric modulator, presumably through the canonical benzodiazepine site in the ECD (Fig. 4a, c, d). At higher concentrations, especially in the presence of flumazenil, DMCM is thought to interact with additional sites in the TMD, where it can potentiate the receptor (Fig. 4b, c, d)[15–18]. We took a strategy like that for the zolpidem complex to obtain the cryo-EM structure of the

**Fig. 4 | DMCM activity and binding sites. a–c** Representative patch-clamp electrophysiology recordings from the cryo-EM construct from at least three independent cells. **a** DMCM at low concentration (1 μM) inhibits GABA response and flumazenil (3 μM) blocks GABA$_A$ receptor inhibition by DMCM (1 μM). **b** DMCM at 10 μM loses its negative modulation activity and potentiates the GABA response in the presence of flumazenil. **c** Negative modulation activity of DMCM attenuates as its concentration increases from 1 μM to 100 μM. **d** Statistical analysis from recordings as in **a–c**. Bar graph shows mean responses with standard deviation for the fractional DMCM modulation at varied concentrations and with co-application of flumazenil. Two-sided Welch's *t*-test was used. A *p*-value of ≤0.05 was considered statistically significant. \*\**p* = 0.0019; \*\*\**p* = 0.0008; \*\*\*\**p* < 0.0001. Replicated numbers from individual cells are shown in the graph. **e** Atomic model of ECD

viewed from synaptic side; yellow box indicates benzodiazepine binding site bound with DMCM (spheres). **f, g** Detailed architecture of DMCM binding pocket boxed in **e**. **f** Synaptic view as in **e**. **g** Side view of the DMCM binding pocket with loop C backbone hidden for clarity. **h** Atomic model of TMD viewed down channel axis from synaptic side; yellow boxes highlight two DMCM binding sites with ligands shown as spheres. **i, j** DMCM binding site details at TMD β/α interface. **i** Side view of the DMCM binding pocket. **j** Synaptic view as in **h**. H-bonds indicated with dashed lines. **k** Two-dimensional free-energy profiles for DMCM binding to the ECD site calculated from funnel metadynamics. The white dots represent the ligand position in the resolved structure. **l** Representative configuration extracted from MD simulations corresponding to the energy minima illustrating the occlusion of DMCM from the H102R receptor. Source data are provided as a Source Data file.

GABA$_A$ receptor in complex with GABA plus DMCM at 2.9 Å overall resolution (Methods, Supplementary Fig. 3). This dataset revealed three DMCM binding sites in total, one in the ECD and two in the TMD. These DMCM sites are shared with zolpidem and help define a structural basis for divergent modulation mechanisms at the benzodiazepine site (Fig. 4e, h).

At the extracellular benzodiazepine site, we were able to precisely position the ligand due to its remarkably clear density (Fig. 4e–g, Supplementary Fig. 4b, c). Its roughly planar structure sits parallel to Loop C, with its dimethoxy head facing the principal (α subunit) side and its ester tail pointing to the complementary (γ subunit) side (Fig. 4f, g). The molecule's β-carboline core is sandwiched between two aromatic residues α1-Y210 and γ2-F77, forming π-stacking interactions. Mutating Y210 to phenylalanine reduces DMCM affinity ~8-fold, suggesting the hydroxyl group contributes to the receptor-ligand interactions as well[29]. Replacement of F77 by either aromatic or non-aromatic residues all resulted in a large loss in ligand stability[28]. Notably, mutations such as F77I have also been observed to trigger potentiating effects of DMCM, consistent with distinct negative and positive modulatory roles of ECD and TMD binding, respectively[47,48]. On the same side of the interface, γ2-T142 forms a potential hydrogen bond with the nitrogen on the carboline benzene ring of DMCM, further stabilizing the ligand (Fig. 4f, Supplementary Fig. 4a). In addition, the twin methoxy groups of DMCM are positioned to form hydrogen bonds (3.1 Å and 2.8 Å) with the α1-H102 indole nitrogen (Fig. 4f, g, Supplementary Fig. 4a). Substitution of this histidine with an arginine has been shown to switch the ligand from a negative allosteric modulator to a positive allosteric modulator[44]. As for zolpidem, we performed funnel metadynamics simulations to study the impact of the H102R mutation on DMCM binding (Fig. 4k). The free-energy surface indicated that introducing arginine at this position expels DMCM from the ECD site, such that the ligand should bind only at the presumably potentiating TMD sites (Fig. 4l).

In the TMD, DMCM binds in the inter-subunit β2-M3 and α1-M1 interface, with its β-carboline ring system sandwiched between β2-F289 and α1-P233 and its methoxy groups orienting away from the channel axis (Fig. 4h–j, Supplementary Fig. 4d–f). Despite improved local signal from additional focused 3D classification, densities for DMCM at both β2/α1 interfaces in the TMD were not as clear as in the ECD (Fig. 4h–j, Supplementary Fig. 4f, i). Together with the near symmetric geometry of the molecule, it was challenging to determine a single orientation from the density map alone. Therefore, we initially modeled the ligand in two poses, one with its ester tail pointing inward (Fig. 4i, j) and another flipped 180° with its ester pointing to the extracellular side. To assess the stability of these two possibilities, we initiated parallel atomistic MD simulations with DMCM in each configuration in each of the two TMD sites. DMCM preferred Configuration 1 at the β2(A)-α1(B) interface but remained stably bound regardless of initial configuration at the β2(C)-α1(D) interface (Supplementary Fig. 4j, k). To assess the reasoning for this difference in stability, we calculated the ratio of residue interactions stabilizing DMCM at the two interfaces (Supplementary Fig. 4j, k), with a

prominent difference being the aromatic F289 residue. Notably, the sidechain of this β2-F289 adopted opposite orientations between the two β2/α1 sites, resulting in slightly different binding modes for DMCM (Supplementary Fig. 4g, h). These findings allowed us to propose a reasonable configuration for DMCM at both sites. They also suggest the two sites are not entirely equivalent due to their differing positions relative to the comparatively dynamic γ2 subunit.

## Structures and simulations suggest a mechanism for DMCM negative modulation

Several modulator classes exert differing effects on GABA$_A$ receptors through the same benzodiazepine site in the ECD. A major interest of the current study was to compare these complexes: zolpidem as a reference positive allosteric modulator, chemically distinct from classical benzodiazepines, and DMCM as a modulator with concentration-dependent negative or positive allosteric effects (Supplementary Fig. 5). The pore conformations in the zolpidem and DMCM complex structures are in a non-conducting, desensitized state, with a closed hydrophobic gate at the base of the pore at the level of the −2' side chains (Supplementary Fig. 5g–i). It is not surprising that DMCM, as a negative allosteric modulator, stabilizes the receptor in a conformation similar to the positive modulator bound structures. Given the DMCM concentration used for the EM sample preparation, the ligand binds at both ECD and TMD sites, functioning as a positive modulator through the TMD and potentiating the GABA response. A DMCM structure with only the ECD site occupied would be helpful for dissecting the drug's biphasic modulation mechanism. However, saturating the ECD site with DMCM while avoiding any TMD site occupancy, without mutagenesis, is likely not possible in an equilibrium structural biology experiment. Computational experiments to simulate selective site occupancy provide a window into the mechanism of ECD-site based inhibition.

To quantify the effects of zolpidem and DMCM on the overall receptor state, we superimposed the complexes reported here with eight previously resolved structures and performed principal-component analysis in Cartesian space on the ECD and TMD of the protein. The six structures bound with GABA alone or GABA plus modulators (diazepam, flumazenil, phenobarbital, etomidate, and propofol) adopt desensitized conformations, with a closed gate at the base of the pore at the level of the −2' side chains. The pore conformations in the presence of GABA plus picrotoxin, and the competitive antagonist bicuculline, contrast with these desensitized states. Picrotoxin, in the presence of GABA, adopts an intermediate state between desensitized and resting, where the ECD adopts a compact agonist-bound conformation while the TMD adopts a more resting-like conformation with the 9' gate partially closed. Bicuculline stabilizes a closed, resting-like state of the pore, with a gate at the 9' position. In comparing the full protein or isolated ECDs and TMDs from various structures (Fig. 5a, Supplementary Fig. 6a, b), the complex with zolpidem clustered with diazepam and other positive modulators, while the complex with DMCM better matched the benzodiazepine antagonist flumazenil. More precisely, DMCM binding was associated

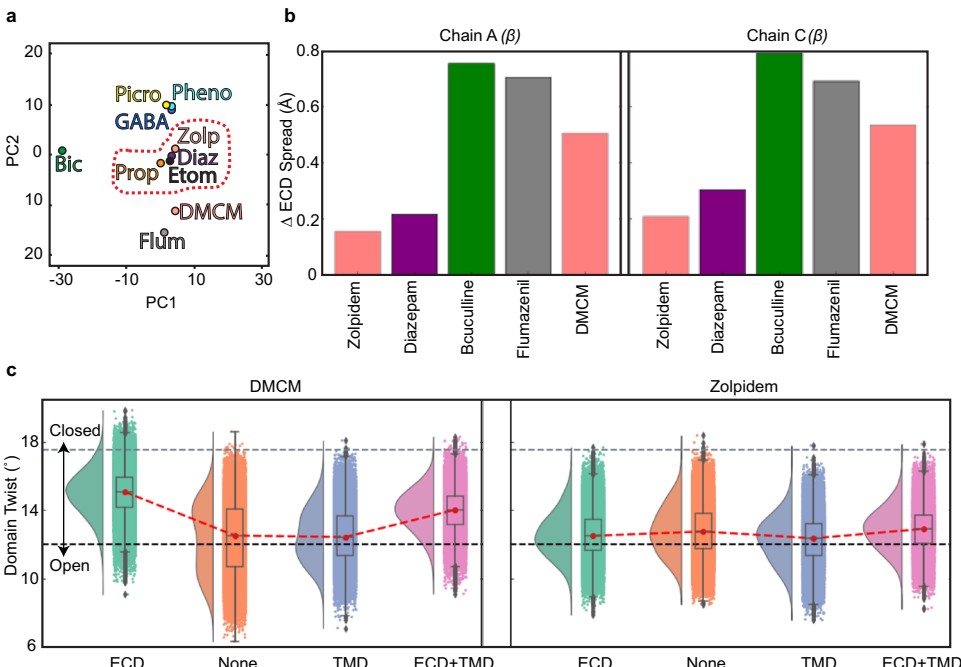

**Fig. 5 | Simulations of benzodiazepine-site modulators. a** Dominant principal components of the ECDs of the zolpidem and DMCM structures together with the 8 previously resolved α1β2γ2 receptor structures. The cluster formed by the PAMs zolpidem, etomidate, propofol and diazepam is illustrated with a dotted line. **b** The ECD spread of β2 subunits (chains A and C) within structures resolved with PAMs/NAMs bound, calculated as the distance between the subunit's Cα atoms center-of-mass (COM) and that of the entire structure's ECD. The spread values are normalized relative to the spread within the structure resolved with GABA alone structure (PDB 6X3Z). **c** Raincloud plots quantifying the relative orientations of the extracellular and transmembrane domains of the β2 subunits from MD simulations with varying ligand binding. Probability distributions are plotted on the left and raw data (n = 48000; 8000 frames each from 2 subunits and 3 simulations) within boxplots are on the right. Boxes span the interquartile range (25th to 75th percentile), whiskers extend up to 1.5 times and outliers are plotted as dots. The median values from each simulation condition are joined by red dotted lines to illustrate differences between them. The domain twist values from resolved resting (PDB 4NPQ) and activated (PDB 4HFI) states of the model pentameric channel GLIC are shown as gray and black dotted lines respectively. Source data are provided as a Source Data file.

with an expansion of the ECD radius across all five subunits relative to the structure with GABA alone (Fig. 5b and Supplementary Fig. 6c). Such an expansion was also observed in the context of flumazenil, and in β subunits bound to the competitive-GABA antagonist bicuculline (Fig. 5b, Supplementary Fig. 6c). Complexes with positive allosteric modulators were associated with less expanded ECDs, along with the α1 and γ2 subunits of the bicuculline complex, which were not bound to inhibitors (Supplementary Fig. 6c). More generally, expansion of the ECD radius (also referred to as blooming) is a characteristic feature of channel closure in multiple pentameric ligand-gated ion channels[49–51], consistent with an opposing effect of DMCM, flumazenil, and bicuculline on potentiation or activation.

To further dissect the role of DMCM binding to ECD and TMD sites on local conformation, we ran molecular dynamics simulations of the DMCM complex with it bound in all sites, or with DMCM removed from either the ECD or TMD sites or both. The presence vs. absence of the ligand in different classes of sites did not dramatically affect local conformation, with the ECD displaying a stable rmsd of ~2 Å in all simulations. However, varying the occupancy of DMCM affected the relative orientation of the extracellular and transmembrane domains of the β2 subunits, quantified as a dihedral of four points (subunit ECD Cα center of mass (COM), entire ECD Cα COM, entire TMD Cα COM, and subunit TMD Cα COM). Simulations with DMCM bound only in the extracellular site were associated with a relatively large rotational angle (~15°) of each subunit's ECD versus TMD (Fig. 5c, Supplementary Fig. 6d). Simulations with DMCM bound only in the TMD, or with no modulator present, typically untwisted to adopt a smaller rotational angle (~13°), while simulations with DMCM in both sites were twisted to an intermediate degree. These rotational angles approached those observed in resting and activated-state structures, respectively, of a

model pentameric ligand-gated ion channel (Fig. 5c, dashed lines); indeed, a comparable decrease in relative domain twist has been proposed as a key initiating step in receptor activation[51]. Simulations of the zolpidem complex showed no such dependence on ligand binding, exhibiting a low twist angle (Fig. 5c, ~13°), consistent with a stable positively modulated state. Comparison of the binding mode of DMCM with other benzodiazepine site modulators as well as the antagonist bicuculline suggest a bulky and rigid chemical core of the ligand, and extensive packing against the complementary subunit, may act in concert to interfere with contraction of the ECD necessary for channel activation, and thereby define important components of a negative modulator (Supplementary Fig. 5a–f). Further elucidation of general rules that determine PAM vs. NAM activity is of great interest and will be challenging.

Taken together, these results suggest a mechanistic model for concentration-dependent effects of DMCM, and for allosterically coupled changes in the ECD and TMD in GABA$_A$-receptor gating (Fig. 6). In the presence of GABA, receptors are in an equilibrium between a resting state with an expanded, twisted ECD, and an activated state with a contracted, untwisted ECD. Binding of zolpidem to ECD and TMD sites does not substantially alter this pattern but shifts the equilibrium to favor the activated over resting state. Conversely, DMCM binding in its extracellular site opposes contraction and untwisting of the ECD, shifting the equilibrium toward the resting state. Additional DMCM binding in its transmembrane sites, as observed in the cryo-EM structure, shifts the equilibrium back toward TMD activation, despite a relatively expanded state of the ECD.

In this study, we have contrasted mechanisms of a benzodiazepine-site potentiator with an inhibitor. The Z-drug bound GABA$_A$ receptor structure provides important information on how the

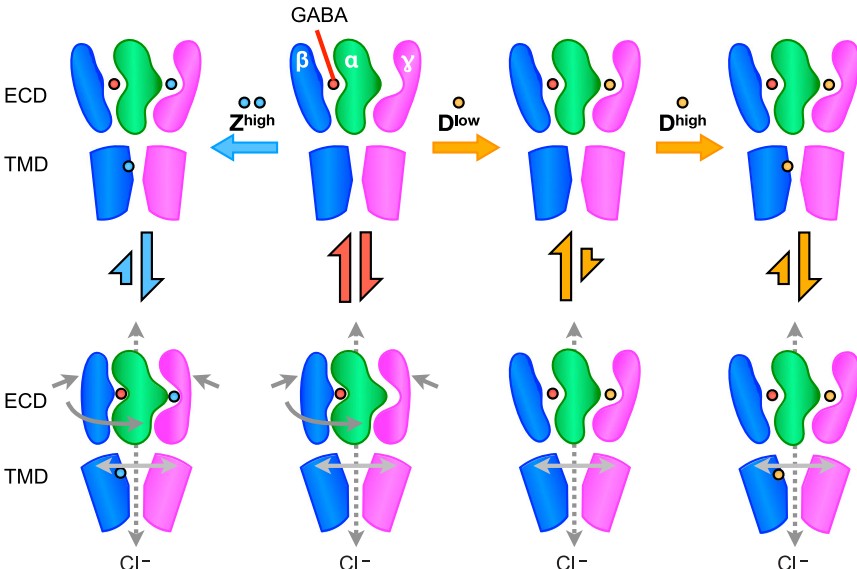

**Fig. 6 | Proposed mechanism of DMCM/Zolpidem activity.** Upon binding of the orthosteric agonist GABA, the structure transitions to an open conformation through a motion that involves the ECD's contraction and rotation, which then triggers the opening of the closed activation gate. This transition to the open conformation is further stabilized by the binding of PAM zolpidem to both the ECD and TMD domains. The NAM DMCM at low concentrations binds to the ECD and stabilizes the twisted and expanded ECD conformation to preclude pore opening. However, at high concentrations, DMCM binds at both ECD and TMD sites to facilitate channel opening despite a twisted and expanded ECD conformation. ECD and TMD motions associated with opening motion are illustrated as dark and light gray arrows, respectively. Although open-pore structures have yet to be captured for the α-β-γ GABA$_A$ receptor, they are expected to transition rapidly to the desensitized states captured by cryo-EM in this and previous work. The distal α and β subunits are hidden in the figures for clarity, along with the proximal α-subunit TMD to better depict changes in the pore radius. D$^{low}$ indicates DMCM at low μM concentrations produces predominantly inhibitory effects. D$^{high}$ indicates DMCM at high μM concentrations produces predominantly potentiating effects.

most prescribed hypnotic works and gives direct structural insights into the drug's selectivity. The structural and computational results complement earlier findings on differential effects at ECD vs. TMD sites for DMCM. The DMCM biphasic switch in activity results from a lower affinity potentiating action through the TMD. MD simulations help resolve ambiguities in structural modeling and build off static snapshots to suggest how DMCM binding in the classical benzodiazepine site inhibits GABA activation. Specifically, DMCM binding in the ECD benzodiazepine site acts like a wedge to inhibit the compaction of the ECD by GABA necessary to open the channel. Together, our work provides new insights into the allosteric modulation mechanisms of the GABA$_A$ receptor.

## Methods
### Protein expression and purification
The heteromeric human α1β2γ2 GABA$_A$ receptor was expressed using a tri-cistronic construct as previously described[20,52]. Briefly, the genes of three subunits were placed in an order of β2-γ2-α1 in the pEZT-BM expression vector and a 22 amino acid long P2A self-cleaving peptide was inserted between subunits[53]. In the EM construct, the M3-M4 loop of each subunit was truncated and replaced by a short linker, SQPARAA[21]. A twin strep-tag was added to the N-terminus of the γ2 subunit for affinity purification. The production and titration of bacmam virus were as described for the α4β2 nicotinic receptor[52]. A total of 4.8 L HEK293S GnTI⁻ suspension cells (ATCC CRL-3022) were transduced a multiplicity of infection of 0.5 and the receptors were expressed at 30 °C with 8% CO$_2$. At the time of transduction, sodium butyrate (Sigma-Aldrich) was added to 2 mM to boost receptor expression. Cells were harvested after 72 h by centrifugation and resuspended in 150 mM NaCl, 20 mM Tris, pH 7.4 (TBS buffer), 1 mM phenylmethanesulfonyl fluoride (PMSF; Sigma-Aldrich), 2 mM GABA (Sigma-Aldrich), and the target ligands: 10 μM zolpidem (Sigma-Aldrich) or 1 μM 3-carbomethoxy-4-ethyl-6,7-dimethoxy-carboline (DMCM, Tocris Bioscience). Cells were disrupted using an Avestin

Emulsiflex, followed by centrifugation at 10,000 g for 20 min. The supernatants containing the cell membranes were centrifuged again at 186,000 g for 2 h. Membrane pellets were homogenized and solubilized at 4 °C for 1 h in a solution containing TBS buffer, 40 mM n-dodecyl-β-maltoside (DDM, Anatrace), 1 mM PMSF and ligands. Insoluble material was removed by centrifugation at 186,000 g for 40 min. The α1β2γ2 GABA$_A$ receptors were purified using Strep-Tactin XT Superflow affinity resin (IBA-GmbH) and eluted in TBS buffer containing 0.01 % (w/v) porcine brain polar lipids (Avanti), 2 mM DDM, ligands (10 μM zolpidem or 1 μM DMCM) and 50 mM biotin (Sigma-Aldrich).

### Nanodisc reconstitution
The plasmid for saposin A expression was a gift from Salipro Biotech AB. The protocol for α1β2γ2 receptor nanodisc reconstitution was modified from the methods in Lyons et al. as described previously[20,54]. The concentrated receptors were pre-mixed with porcine brain polar lipids (Avanti) at room temperature for 10 min. Subsequently, saposin was added to the mixture and incubated for 2 min. The molar ratio of receptor, lipids and saposin was 1:230:30. The mixture solution was diluted ~10-fold with TBS buffer to initiate reconstitution. Removal of detergent was achieved by addition of Bio-Beads SM-2 (Bio-Rad) at a maximum concentration of 200 mg/mL while the sample was rotating at 4 °C. Bio-Beads were removed by centrifugation after overnight incubation. The receptor-lipid-saposin complexes were collected and concentrated to <500 μL for size exclusion chromatography.

### Cryo-EM sample preparation
Purified receptor-nanodisc complexes were mixed with IF4 Fab fragment in a 4:1 (w/w) ratio, rotating for 20 min at 4 °C[21]. Further purification was performed by injecting the sample over a Superose 6 Increase 10/300 GL column (Cytiva) equilibrated in TBS buffer supplemented with ligands. Sample quality was assessed by size exclusion chromatography, monitoring tryptophan fluorescence. The peak

fractions were collected and concentrated to an A280 of 7-9. The final protein sample was supplemented with zolpidem or DMCM with a final concentration of 100 μM or 500 μM, respectively.

To induce random particle orientations in grid holes, 0.5 mM fluorinated Fos-Choline-8 (Anatrace) was mixed with the receptor sample immediately before EM grid freezing. The final protein product (3 μL) was applied to glow-discharged gold or copper R1.2/1.3 200 mesh holey carbon grids (Quantifoil) before blotting (3.5 s) at 100% humidity and 4 °C, then plunge-frozen into liquid ethane using a Vitrobot Mark IV (FEI).

## Cryo-EM image collection and processing
Each dataset was collected over 48 h on a FEI Titan Krios Microscope (Thermo Fisher) operated at 300 kV and equipped with a K3 direct electron detector (Gatan) and a GIF quantum energy filter (20 eV) (Gatan). Extended Form 1 lists dataset details. The datasets were processed using the general workflow in RELION 3.1[46] as follows: dose-fractionated images (movies) were gain normalized, 2x Fourier binned, aligned, dose-weighted and summed using MotionCor2[55]. GCTF[56] was used to estimate the contrast transfer function (CTF). For the zolpidem dataset collected at the UTSW north campus EM facility, several hundred particles were manually picked in Relion and then subjected to 2D classification to generate a template for autopicking. The DMCM structure dataset was collected at the PNCC on Krios 4. Particle picking was performed using crYOLO[57] and picked particles were subjected to reference-free 2D classification in Relion 3.1. After 3 rounds of reference-free 2D classification, classes exhibiting clear GABAA receptor shapes were selected and subjected to 3D classification using an ab initio model, generated from 3,000-5,000 good particles in RELION. 3D classes with high-resolution features were pooled for 3D refinement. Because of the heterogeneity we observed in the transmembrane domain (TMD) of the γ2 subunit in both datasets, γ subunit TMD focused classification was performed after subtracting the signal from the rest of the receptor and nanodisc. Particles from the best classes were selected for particle polishing, CTF refinement and 3D refinement and B factor sharpening to generate the final maps. Local resolution was estimated with ResMap[58].

## Model building, refinement and validation
The model from a GABAA Receptor-Fab complex bound to GABA + diazepam (RCSB PDB: 6X3X) was used as a starting model after removing ligands. It was docked into the density map using UCSF Chimera[59] and manual adjustment was performed in Coot[60]. GABA and zolpidem or DMCM were fitted into their binding sites based on features in the density map and the local chemical environment. After manual building in Coot, global real space coordinate and atomic B-factor refinement with stereochemical restraints were performed in Phenix[61]. The quality of final models was assessed using Phenix and Molprobity[62]. Schematic interaction analysis of the bound ligands was performed using Ligplot⁺[63]. Pore radius profiles were analyzed using Hole2[64]. Sequence alignments were made using PROMAL3D[65]. Structural figures were generated with UCSF-Chimera and Pymol (Schrodinger, LLC). Structural biology software packages were compiled by SBGrid[66].

## Electrophysiology
The tri-cistronic pEZT construct was transiently transfected to adherent HEK293S GnTI⁻ cells for performing whole cell voltage-clamp recordings. Each well in a 12-well dish of cells was transfected with 0.2 μg of the plasmid DNA 1–2 days before recording. On the day of recording, cells were re-plated onto a 35 mm dish and washed with bath solution (in mM): 140 NaCl, 2.4 KCl, 4 MgCl₂, 4 CaCl₂, 10 HEPES pH 7.3, and 10 glucose. Borosilicate pipettes were pulled and polished to an initial resistance of 2–4 MΩ. The pipette solution contained (in mM): 150 CsCl, 10 NaCl, 10 EGTA, and 20 HEPES pH 7.3. Cells were clamped at

−75 mV. The recordings were made with an Axopatch 200B amplifier, sampled at 5 kHz, and low pass filtered at 2 kHz using a Digidata 1440 A (Molecular Devices) and analyzed with pClamp 10 software (Molecular Devices). The GABA, zolpidem, DMCM and flumazenil solutions were prepared in bath solution. A gravity driven RSC-200 rapid solution changer (Bio-Logic) was used for solution exchange. The experiments were repeated for at least 3 times from three different cells. All statistical analyses were conducted using GraphPad Prism 9.1.2.

## Statistical analysis
Statistical analysis was performed using GraphPad Prism 9.2.0 software (GraphPad software, Inc, La Jolla, CA). Data are expressed as means ± S.D of at least three recordings from independent cells. Two-tailed Welch's t-test was used. A p-value of ≤0.05 was considered statistically significant.

## Molecular dynamics
The protein structures were embedded in a POPC membrane of dimension 15x15 nm, solvated in TIP3P and 0.15 M NaCl using Charmm-GUI[67]. The proteins, lipids and ligands were described using the Amber99SB-ILDN[68], Slipids[69] and GAFF2[70] force fields respectively. After energy minimization and equilibration, three replicates of each system were simulated for 1 μs using Gromacs 2019[71] and a timestep of 2 fs. Long-range electrostatic interactions were calculated using the particle mesh Ewald method and hydrogen-bond length were constrained using LINCS[72]. Pressure and temperature were maintained through the use of the Parrinello-Rahman[73] barostat and v-rescale[74] thermostat respectively. The ECD-TMD twist angles were calculated using MDAnalysis[75] scripts as an average over the two β subunits from the final 500 ns of all replicates.

## Funnel metadynamics simulations
Metadynamics simulations promote sampling by discouraging the return to visited regions of the Collective-Variable (CV) space through the addition of a history-dependent bias. However, within protein-ligand simulations, metadynamics simulations are unable to sample a statistically significant number of binding/unbinding events due to the large number of unbound configurations in the solvent phase. This limitation can be circumvented through the introduction of a funnel-shaped restraint where the conical section allows sampling of bound states, and the cylindrical section restricts unbound phase-space exploration (Supplementary Fig. 2a). In this study, the computational cost of funnel metadynamics necessitated the use of only the ECD of the α-γ benzodiazepine binding interface. The allosteric effects of the TMD and other subunits on drug are thus precluded from the study.

Funnel metadynamics simulations were performed using the methodology of Comitani et al.[76]. The α/γ ECD subunits were extracted from the cryo-EM structure together with the bound ligand and placed in dodecahedron solvent box with 0.15 M NaCl. The axis of the funnel was described as a vector from the center of mass (COM) of 7 residues surrounding the ligand (α-F100, α-H102, α-Y160, α-Y210, γ-Y58, γ-F77, γ-T142) to the COM of two residues making up the loop C turn (α-S206, α-T207). The cone angle was set to 20° and transitioned to a cylindrical restraint of radius 1 Å at a distance of 2.5 nm.

To compensate for the non-inclusion of the other subunits, the Cα atoms of five initial and final ECD residues were fixed using harmonic restraints of 1000 kJ/mol/nm². Preliminary metadynamics simulations suggested that the thermal fluctuations of loop C were insufficient to allow a statistically significant number of ligand binding/unbinding events. A two-dimensional metadynamics simulation was thus employed that (i) biased the distance between the COM of the ligand and those of the above mentioned 7 residues and (ii) the projection of loop C along the funnel axis. Gaussians were deposited at intervals of 2 ps with an initial height of 2 kJ/mol and bias factor of 15. The

simulations were considered converged when a significant number of recrossing events between the bound and unbound states. The trajectories of the ligand were remapped using the reweighting algorithm of Bonomi et al. [77] to calculate the 2-dimensional free-energy profiles ligand binding.

## Principal-component analysis

Protein models in complex with zolpidem and DMCM, along with previously reported complexes with propofol, etomidate, diazepam, picrotoxin, bicuculline, phenobarbital, and flumazenil (PDB IDs 6X3T, 6X3V, 6X3X, 6X40, 6X3S, 6X3W, and 6X3U) were aligned onto the structure with GABA alone (PDB ID 6X3Z) using all Cα atoms. Principal components of motion were then calculated in Cartesian space using the Cα coordinates for the ECD (residues equivalent to β2 10-217), TMD (residues equivalent to β2 218-338), or the whole protein in all subunits.

## Reporting summary

Further information on research design is available in the Nature Research Reporting Summary linked to this article.

## Data availability

Atomic model coordinates for GABA + zolpidem and GABA + DMCM-bound structures have been deposited in the Protein Data Bank with accession codes 8DD2 and 8DD3, respectively, and the cryo-EM density maps have been deposited in the Electron Microscopy Data Bank with accession codes EMD-27332 and EMD-27333, respectively. Source data are provided with this paper.

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

## Acknowledgements

We thank D. Legesse and C. Noviello for critical reading of the manuscript. Single-particle cryo-EM grids were screened at the University of Texas Southwestern Medical Center Cryo-Electron Microscopy Facility, which is supported by the CPRIT Core Facility Support Award RP170644; the zolpidem dataset was collected at this facility. A portion of this research was supported by NIH grant U24GM129547 and performed at the PNCC at OHSU and accessed through EMSL (grid.436923.9), a DOE Office of Science User Facility sponsored by the Office of Biological and Environmental Research. The DMCM dataset was collected at the PNCC. A portion of this research was supported by NIH grant U24GM129547 and performed at the PNCC at OHSU and accessed through EMSL (grid.436923.9), a DOE Office of Science User Facility sponsored by the Office of Biological and Environmental Research. S.Z. acknowledges a postdoctoral fellowship from the American Heart Association. A.S. was supported by Marie Sklodowska-Curie grant 898762, and E.L. and R.J.H. by grants from the Swedish Research Council (2017-04641, 2019-02433) and Swedish e-Science Research Center. This project was supported by grants from the NIH (DA047325) and the Welch Foundation (I-1812) to R.E.H.

## Author contributions

S.Z. performed the sample preparation, EM data collection for the zolpidem complex, and data processing for cryo-EM, structural analysis and drafted the manuscript with R.E.H. J.T. performed the electrophysiology and revised the manuscript. R.E.H. assisted in structural analysis and model validation and directed the project. A.S. performed the simulations and drafted MD sections of the manuscript. A.S., R.J.H., E.L., S.Z. and R.E.H. revised the manuscript with input from all other authors.

## Competing interests

The authors declare no competing interests.
