## [Peer Review File · Nature Communications]

Structural and dynamic mechanisms of GABAA receptor modulators with opposing activitiesREVIEWER COMMENTS

Reviewer #1 (Remarks to the Author):

Benzodiazepines are a major class of drugs used for sedation, muscle relaxation and anxiolysis. They target the GABAA receptors in the brain, and the Hibbs laboratory already published the cryo-EM structure of the main synaptic GABAA receptor subtype in complex with the prototypic Benzodiazepine diazepam. They found a high affinity binding site at the extracellular alpha/gamma interface, and three low affinity binding sites at the gamma/beta and beta alpha interfaces. In the present paper, they extend the structural analysis to two additional allosteric modulators, zoldipen, that is a new generation positive allosteric modulator (PAM) used for sedation, and DMCM that acts as a negative allosteric modulator (NAM) at low concentration, but as a PAM at high concentration. The structures were solved in complex with each modulator and GABA, and show a “desensitized conformation” of the ion channel. Both structures show, as expected, binding at the extracellular alpha/gamma interface, and in the transmembrane domain only at the two alpha/beta interface. The structures allow dissecting the molecular details of the interactions the ligands make with the various amino acids, especially at the high affinity extracellular site. Data are interpreted in the framework of the published mutagenesis and electrophysiological data, and specific mutations are explored by molecular dynamics simulations named “funnel metadynamics”. Overall, data give a quite consistent picture of the key residues contributing to ligand binding and specificity among subtypes, a feature that is key concerning the therapeutic and side effect of zoldipen and other benzodiazepines.

The two structures, together with 6 previous structures solved by the Hibbs lab in complex with various ligands (bicuculine, diazepam, various general anesthetics and picrotoxin), are further analyzed by principal component analysis and molecular dynamics. Principal component analysis suggests that the structure in complex with zoldipen better matches that with PAMs, while that of DMCM better match that with the NAM flumazenil. In addition, molecular dynamic simulations of each structure, or with DMCM bound only to the ECD or only to the TMD, suggest a mechanistic model with an expanded and twist ECD for the resting state. MD shows that NAMs tend to drive the structure toward the resting state, while PAMs, as well as DMCM bound to the TMD, tend to drive the structure toward the active/desensitized state.

Altogether, the structures are of good quality and give important insights not only about the structural pharmacology of important therapeutic drugs, but also about the mechanisms underlying their functional effect through carefully designed simulations. The work is globally clearly presented and well conducted, but I have two main concerns:

1/ A key aspect of the work is the interpretation of the high-resolution structural data in relation to functional data. However, the functional data presented in the paper related to the engineered construct used in Cryo-EM are hardly convincing. They are presented in supplementary figure 2 and figure 4. In each case, only one trace is presented with no statistical analysis. In addition, in

supplementary figure 2, the inhibition of zolpidem potentiation by FLM is partial and not convincing. Likewise, in supplementary figure 4 the effect of DMCM appears quite weak (trace b).

Another concern is the effect of the $\alpha 1$ -H102R mutation which is proposed to be key in the selectivity of zolpidem among subtypes. The mutation is explored by MD simulation, but not by electrophysiology. It would be thus important to perform this experiment.

2/ The second concern is related to the presentation of the structures and of the computational data. It is not clear to me in which conformation the two structures are, and how they compare to the previously published structures. The data provided in the figures are important about specific twisting or blooming motion, but what about the overall conformation? What is the extent of twist and blooming motion? Author should present the global structures in supplementary material, showing superimposition, rmsd analysis, or any relevant representation that can give the reader an idea of the changes in conformation elicited by the two modulators as compared to the previously published structures.

Minor point:

The funnel metadynamics approach is not explained in the text. It should be presented in more details, including presentation of the limitation of the approach.

Reviewer #2 (Remarks to the Author):

This work presents a structural and dynamical analysis of a GABA receptor ($\alpha 1\beta 2\gamma 2$) in complex with either zolpidem or a β -carboline compound; methyl-6,7-dimethoxy-4-ethyl- β -carboline-3-carboxylate known as DMCM for short. Zolpidem is of interest as this is representative of the latest generation of sedatives, but the DMCM is an interesting example of a compound that binds to the same site (in the ECD) yet exerts negative rather than positive allosteric modulator effects. I found the work well-presented and very easy (and enjoyable) to read.

The first part of the paper deals with zolpidem and the basis of its selectivity at the $\alpha 1\gamma 2$ interface. The cryo-EM work is nicely supported by MD and funnel meta dynamics nicely confirms the role of a key histidine.

The second part tries to address the fundamental question of what makes a NAM a NAM. To get at this, the authors use a series of simulations, but its not very clear exactly what was actually run - they say “ran molecular dynamics simulations of the DMCM complex with all ligands bound, or with DMCM removed from some or all its ECD or TMD sites” It looks from the figures like nothing bound, DMCM bound everywhere, DMCM just at the TMD and DMCM just at the END, but I am not entirely sure? Also - how many sims were performed for each etc?

The authors then say that the ECD RMSD was still very low but that there was a pattern of tertiary blooming consistent with closure of this family of receptors. This part of the work is very interesting, but the reader is still left wondering what it is about DMCM that makes the structure “bloom” and it feels like the authors kind of ran out of ideas at this point. One might think that NAMs are slightly bigger than PAMs for example – is this true? (I don’t know) – or do they make a specific interaction? – In other words, what is that causes the change in the relative positioning of the ECD with respect to the TMD? I feel like this is super-important because really what we want is to have rules so we are able to say – “if you want a PAM, don’t have X or Y etc”. It feels like the authors are tantalizingly close here, and it at the very least warrants a bit more in-depth discussion even if they cannot resolve this.

Minor points

Fig 2. It says H-bonds are indicated with a dashed line. In my version of the pdf as least, I could only (just) see a dashed line in panel b between T207 and one of the imidazopyridine nitrogens (with what looks like poor h-bond geometry but could just be the view). Would be useful to make the h-bonds more obvious. It might be better to slightly move the view because the S205 h-bond talked about in the text is also not visible. And according to Supplemental 2e, these are the only two hydrogen bonds present.

Fig. 3. What are the coordinates on the X-Y axis?

Reviewer #3 (Remarks to the Author):

Zhu et al report cryo- EM structures of zolpidem- and DMCM- bound a1b2g2 GABA-A receptor constructs. The collaborators from Sweden contributed MD simulations of various ligand bound states,

some of which are not accessible to the experiment: namely receptors with only some of the observed binding sites occupied. With the total body of data, a mechanism for negative and positive allosteric modulation by DMCM is put together from the static structures and the simulations. The subtype selectivity of zolpidem is examined in the light of the observed binding mode. In total, this is a very valuable study which adds to the knowledge of the mechanisms governing allosteric modulation. The results thus can be expected to pave the way towards a more rational development of compounds with desired profiles, as well as to future improved drugs.

There are some points of concern that should be addressed prior to publication as listed point by point:

The major concerns:

The functional data that was obtained with the used constructs and ligands is only presented in the supplementary figures, and mentioned “in passing”. It does, however, add value to the study, and raises questions that deserve to be addressed. The overall balance between the presentation of results obtained from the structures, the MD simulations and electrophysiology should be reconsidered, giving adequate room to the functional data in the MS and potentially compacting long, rather descriptive passages that deal with aspects of the presented structures.

The questions raised by the functional data in detail:

Concern 1: Supplementary Fig 2a and accompanying MS text:

In the methods it is stated that experiments were done with $n \geq 3$. Where is the data beyond a single representative trace? The figure legend for 2a does not give the precise n for the representative observations. Mean values for the reference currents and the modulation are not reported. All data resulting from experiments described in the methods should be integrated. Line 100 (where the Suppl. figure 2a is called) states “native-like response” – this suggests a comparison with a more native set of constructs. Plz reword to reflect that the data corresponds with published observations, if this is what is meant. In panel a, the 4th trace shows the observed current in presence of 5 microMol GABA, 1 microMol Zolp and 3 microM FLM. This is interpreted as competitive inhibition on Zolp modulation by FLM – this should be demonstrated by a quantitative comparison between traces 2 and 4, using all datasets. Moreover: The samples for structure determination were prepared in presence of 10 microMol Zolp, and the functional effect of the presence of Zolp at the TMD sites is described as “remains unclear” in line 134. Perhaps an experiment similar to the one shown in the fourth trace but with higher Zolp concentration would help here to either rule out, or demonstrate that zolpidem modulates at higher concentrations in the presence of FLM. In Supplementary Figure 4a, the corresponding experiment with DMCM was performed at 10 microM DMCM.

Concern 2: Supplementary Figure 4a and accompanying MS text:

The same issues as discussed for zolpidem apply to this dataset as well: Panels a and b provide representative traces, but the quantitative data analysis is missing. The legend states “representative results from $n=3$ recordings from different cells” – is a single representative trace per experiment shown? (Or averaged traces, which seems the less plausible interpretation of the legend text?) Please

provide a summary of analyzed data, with all relevant comparisons (reference currents compared to the co-application experiments). Moreover: Panel b, second trace: It looks like the application was stopped during the rising phase, without reaching peak current. Another issue is the statement in line 226 “as well as directly activate the receptor (Supplementary Fig 4b)”: No experiment in the absence of GABA is displayed in Supplementary Fig 4b, please provide this data, or rewrite accordingly.

Related to the issues with the presentation of functional data for DMCM: Line 226 indicates “at low concentrations acts as an inverse agonist, or negative allosteric modulator” – presumably the intention is to state that it acts as negative allosteric modulator (also historically termed benzodiazepine receptor inverse agonist). Please rewrite using clear and IUPHAR conforming terminology. See also related minor point 7 below.

Concern 3: The subtype selectivity of zolpidem is explained by the observed binding mode and the absence or presence of specific ligand- protein interaction. This analysis would greatly benefit from a thorough comparison with the available structure of the alpha5 subunit to shed light on the impact the loop B has on the ability of the ligand to bind with high affinity at a1, a2 and a3, but not at interfaces formed by the so-called zolpidem insensitive subunits a4, a5 and a6 which have a proline in the position homologous with a1-T163 (discussed in lines 174ff). The “more complex” influence that the proline may have can possibly be deduced from the alpha5 subunit’s structure – at least this should be attempted.

Also concerning the determinants of zolpidem selectivity in loop C, the impact of a1-V203 (lines 188-190) seems overstated, the EC50 values in the zolpidem sensitive a2 and a3 – containing receptors are only slightly to the right of those for the a1- case (see e.g. • DOI: 10.1016/j.ejphar.2010.03.015)

Minor points:

1: Abstract, line 26: Based on currently approved medications and their ATC codes, it seems inappropriate to list depression side by side with insomnia and epilepsy – no GABA-A R targeting medication bears an ATC code as antidepressant, but many as anxiolytics or hypnotics, and several as antiepileptics. Use in patients with depression is strictly off label at the time, and preclinical research into this direction has yet to provide compounds suitable for clinical trials as monotherapy in depression. (This point might seem a bit nit-picking, but it should be easy to provide a correct and balanced view of what is actually really approved and for which indication)

2: Line 31: sedative should read sedative- hypnotic, or just hypnotic. The main indication of zolpidem is as sleeping aid, not as sedative.

3: Lines 45 and 52: “anxiety” is not a disorder. The correct term is “anxiety disorders”, while anxiety is a perfectly normal emotional response to threatening stimuli. (Again, this might seem overly picky – but the clinical scientists are, rightfully, irritated by such incorrect use of terminology that sadly is widespread in the preclinical literature.)

4: Lines 68-70: The end of the sentence should read “to enhance GABA- elicited currents or to elicit currents” as it is stated that DMCM can act as allosteric agonist.

5: Lines 145-148: Reference 33 reflects work in rodents. The notion that $\alpha 1$ -containing receptors mediate sedation in humans has been rejected (see doi: 10.1016/j.tips.2012.08.003.) – and still is often re-stated as a never verified myth in countless research articles and reviews.

6: Line 215: Should this read “helping to form”?

7: Lines 226, 249, 279: In most places in the article, the authors use the terms positive/ negative allosteric modulators, but in these places (and possibly elsewhere) they use the outdated terms “inverse agonist/ agonist” which are misleading as they can be confused with orthosteric ligand effects (historically, the full term was “inverse agonist/ agonist of the benzodiazepine receptor”). IUPHAR recommendation should be followed throughout, it can be stated in the introduction that benzodiazepine site PAM/ NAM compounds are often termed “inverse agonist/ agonist of the benzodiazepine receptor” – chiefly in older papers.

8: Line 361: “ligands” is unspecific

9: Figure 5 and legend: The chain labels in panel b are hard to see, read and interpret. Perhaps “chain A” and “chain C” respectively would be better. Panel c: It is a bit puzzling to see that simulations with no ligand (None, orange plots) are quite different for the two panels with a biphasic appearance in the DMCM image, and a monophasic one in the right gallery. Some words of explanation might be helpful.

10: Figure legend for Fig 6, line 691: alpha-beta-gamma should either all be Greek letters, or all spelled out.

11: Figure 6: The cartoon aimed to depict the motions that lead to positive or negative allosteric effects is not particularly illustrative or intuitive, the “expanded” versus “contracted” ECD should be intuitively obvious. Several improvements of this cartoon can be suggested: (a) The legend should clearly explain why only 2 subunits are shown for the TMD, while three depict the ECD. (b) D-low and D-high need to be defined. The left image suggests that zolpidem binds with the same (high) affinity at ECD or TMD sites – or should it state Z-high? (c) Expanded and contracted ECDs should look differently.

12: Supplementary Fig 2f: Amino acid labels to match panel e would be helpful.

13: Same as remark 12, for panel d and c of Supplementary Figure 4.

14: Supplementary Figure 5, amino acid labels in panels a-d would be helpful.

Congrats to all authors, looking forward to seeing this published,

margot ernst

Response to reviewer comments:

We thank the three reviewers for their careful reading, enthusiasm, and constructive critiques. We have added new electrophysiology results to the study and brought these into the main figures, to address consensus reviewer requests. We have also revised the text to be more clear about our mechanistic interpretations, and to be more precise in clinical and pharmacological terminology. These revisions have substantially strengthened the study. We present point-by-point responses to unedited reviewer comments below. Reviewer comments are in bold; our responses are in standard font.

Reviewer #1 (Remarks to the Author):

Benzodiazepines are a major class of drugs used for sedation, muscle relaxation and anxiolysis. They target the GABAA receptors in the brain, and the Hibbs laboratory already published the cryo-EM structure of the main synaptic GABAA receptor subtype in complex with the prototypic Benzodiazepine diazepam. They found a high affinity binding site at the extracellular alpha/gamma interface, and three low affinity binding sites at the gamma/beta and beta alpha interfaces. In the present paper, they extend the structural analysis to two additional allosteric modulators, zolpidem, that is a new generation positive allosteric modulator (PAM) used for sedation, and DMCM that acts as a negative allosteric modulator (NAM) at low concentration, but as a PAM at high concentration. The structures were solved in complex with each modulator and GABA and show a “desensitized conformation” of the ion channel. Both structures show, as expected, binding at the extracellular alpha/gamma interface, and in the transmembrane domain only at the two alpha/beta interface. The structures allow dissecting the molecular details of the interactions the ligands make with the various amino acids, especially at the high affinity extracellular site. Data are interpreted in the framework of the published mutagenesis and electrophysiological data, and specific mutations are explored by molecular dynamics simulations named “funnel metadynamics”. Overall, data give a quite consistent picture of the key residues contributing to ligand binding and specificity among subtypes, a feature that is key concerning the therapeutic and side effect of zolpidem and other benzodiazepines.

The two structures, together with 6 previous structures solved by the Hibbs lab in complex with various ligands (bicuculine, diazepam, various general anesthetics and picrotoxin), are further analyzed by principal component analysis and molecular dynamics. Principal component analysis suggests that the structure in complex with zolpidem better matches that with PAMs, while that of DMCM better match that with the NAM flumazenil. In addition, molecular dynamic simulations of each structure, or with DMCM bound only to the ECD or only to the TMD, suggest a mechanistic model with an expanded and twist ECD for the resting state. MD shows that NAMs tend to drive the structure toward the resting state, while PAMs, as well as DMCM bound to the TMD, tend to drive the structure toward the active/desensitized state.

Altogether, the structures are of good quality and give important insights not only about the structural pharmacology of important therapeutic drugs, but also about the mechanisms underlying their functional effect through carefully designed simulations. The work is globally clearly presented and well conducted, but I have two main concerns:

1/ A key aspect of the work is the interpretation of the high-resolution structural data in relation to functional data. However, the functional data presented in the paper related to the engineered construct used in Cryo-EM are hardly convincing. They are presented in supplementary figure 2 and figure 4. In each case, only one trace is presented with no statistical analysis. In addition, in

supplementary figure 2, the inhibition of zolpidem potentiation by FLM is partial and not convincing. Likewise, in supplementary figure 4 the effect of DMCM appears quite weak (trace b). Another concern is the effect of the alpha1-H102R mutation which is proposed to be key in the selectivity of zolpidem among subtypes. The mutation is explored by MD simulation, but not by electrophysiology. It would be thus important to perform this experiment.

Both reviewers 1 and 3 brought up this excellent point about our electrophysiology data presentation being superficial. We strongly agree and present new results with statistical analysis to address this shortcoming of the original submitted manuscript. We include these data now in the main figures to highlight their importance and make them more easily viewable. An example recording and statistical analysis from several independent cells are now shown as part of Figure 2 and 4. On the topic of FLM inhibition of zolpidem potentiation, now shown in main Fig 2, compared to GABA alone, the current is slightly larger in the presence of GABA + FLM + zolpidem, likely because FLM competes off the ECD zolpidem, but fractional TMD-bound zolpidem could contribute to potentiation. This result mirrors that of FLM competition of diazepam, shown in PMID 32879488, Extended Data Figure 11b. In our new Figure 2b, application of FLM appears to slightly increase GABA currents, however this difference is not statistically significant ($p=0.0901$). By contrast, GABA vs GABA + FLM + zolpidem does reach a statistically significant difference ($p=0.042$). These differences between GABA alone and test conditions are very small compared the GABA vs. zolpidem ($p=0.022$).

Regarding DMCM, we present the new data and analysis in main Figure 4. Through analysis of replicates (independent cells), we now show dose-dependent potentiation by DMCM in the presence of flumazenil. This potentiation is large (nearly double the response from GABA alone). By contrast, the inhibition of GABA currents by DMCM (in the absence of flumazenil) is significant, and dose dependent, but smaller (~25% decrease). A survey of the literature reveals a wide range of levels of inhibition of GABA currents by DMCM, ranging from 25-60% inhibition (PMID 9824848). Our results are, thus, on the low end of those observed by others, which could relate to subunit composition, heterogeneity in subunit composition, and recording conditions.

An important point that we now make clear in the manuscript is that we and others do not consistently see potentiation by DMCM in the absence of FLM. For example, E Costa's group observed that 10 μ M DMCM alone exerts inhibitory effect on the GABA-induced currents (PMID: 2476816). When they co-applied DMCM with flumazenil, they observed potentiation of the currents. They performed the experiments in HEK cells and our results are consistent with their findings. A study from Sergeeva's group examined the effects of a panel of GABA_A receptor modulators on the mouse neurons. They found 100 μ M DMCM still inhibited GABA induced currents (PMID 23799902). Two studies reported that a high concentration DMCM can potentiate GABA responses. However, experimental conditions are quite different from ours. They used *Xenopus* oocytes (PMID 8848011) and frog neurons (PMID 2574062). We now include a new Fig 4c that showing that the NAM effect of DMCM attenuates as the concentration of DMCM increases to 100 μ M.

A study from Peter Seeburg's group examined the role of alpha1-H102 in zolpidem binding and found that mutagenesis to arginine ablated binding (PMID 1346133). We reference this study in the manuscript. They and others found that reverse mutagenesis of the arginine (to histidine) in this position in $\alpha 4$ and $\alpha 6$ subunit did not rescue binding, consistent with our descriptions of H102 not being the principal determinant of sensitivity in the $\alpha 1$ containing receptors like it is for classical benzodiazepines. We feel that further electrophysiology on an $\alpha 1$ H102R mutant would not be more informative than the published binding results.

2/ The second concern is related to the presentation of the structures and of the computational data. It is not clear to me in which conformation the two structures are, and how they compare to the previously published structures. The data provided in the figures are important about specific twisting or blooming motion, but what about the overall conformation? What is the extent of twist and blooming motion? Author should present the global structures in supplementary material, showing superimposition, rmsd analysis, or any relevant representation that can give the reader an idea of the changes in conformation elicited by the two modulators as compared to the previously published structures.

Thank you for raising this point about lack of clarity in the conformational assignment of the structures. Both the zolpidem and DMCM complexes are in desensitized conformations, with the tightest constriction near the base of the pore at the -2' position. We explicitly state this in the revised manuscript at the start of the section titled "Structures and simulations suggest a mechanism for DMCM negative modulation."

In the second paragraph of that section, we said we compared the two structures with the previous 8 structures (from PMID 32879488), but we did not say which conformations these earlier structures are assigned to, which added to confusion. The six structures bound with GABA alone or GABA plus modulators (diazepam, flumazenil, phenobarbital, etomidate and propofol) adopt desensitized conformations, with a closed gate at the base of the pore at the level of the -2' side chains. The pore conformations in the presence of GABA plus picrotoxin, and the competitive antagonist bicuculline, contrast with these desensitized states. Bicuculline stabilizes a closed, resting-like state of the pore, with a gate at the 9' position. Picrotoxin, in the presence of GABA, adopts we called an intermediate state between desensitized and resting, where the ECD adopts a compact agonist-bound conformation while the TMD adopts a more resting-like conformation with the 9' gate partially closed.

We now describe in the text the conformational states of the earlier structures, and which ones are in the same class as the current zolpidem and DMCM complexes. Zolpidem and DMCM are benzodiazepine site modulators, thereby, their structures adopt similar conformational states as diazepam and flumazenil complexes. In our previous paper (PMID 32879488), there are very comprehensive conformational comparisons of the eight structures. Here, we revise our HOLE graph comparing the new ones with GABA alone, GABA+diazepam and GABA+flumazenil, further illustrating the pore conformations of structures bound by different types of benzodiazepine site modulators.

On the topic of the extent of twisting and blooming, in the revised manuscript we discuss the details in lines 311-317 of the main text and present the data in Figure 5b, c. We made figures to attempt to illustrate the differences between the two new structures by superimposition, but the presentation was not clear. Your suggestion to compare rmsd values among several structures was informative and we now include this information as a new Supplementary Fig 6b.

Minor point:

The funnel metadynamics approach is not explained in the text. It should be presented in more details, including presentation of the limitation of the approach.

We include the following statement now in the Methods section:

Metadynamics simulations promote sampling by discouraging the return to visited regions of the

Collective-Variable (CV) space through the addition of a history-dependent bias. However, within protein-ligand simulations, metadynamics simulations are unable to sample a statistically significant number of binding/unbinding events due to the large number of unbound configurations in the solvent phase. This limitation can be circumvented through the introduction of a funnel-shaped restraint where the conical section allows sampling of bound states, and the cylindrical section restricts unbound phase-space exploration (Supplementary Fig. 2a). In this study, the computational cost of funnel metadynamics necessitated the use of only the ECD of the α - γ benzodiazepine binding interface. The allosteric effects of the TMD and other subunits on drug are thus precluded from the study.

Reviewer #2 (Remarks to the Author):

This work presents a structural and dynamical analysis of a GABA receptor ($\alpha 1\beta 2\gamma 2$) in complex with either zolpidem or a β -carboline compound; methyl-6,7-dimethoxy-4-ethyl- β -carboline-3-carboxylate known as DMCM for short. Zolpidem is of interest as this is representative of the latest generation of sedatives, but the DMCM is an interesting example of a compound that binds to the same site (in the ECD) yet exerts negative rather than positive allosteric modulator effects. I found the work well-presented and very easy (and enjoyable) to read.

The first part of the paper deals with zolpidem and the basis of its selectivity at the $\alpha 1\gamma 2$ interface. The cryo-EM work is nicely supported by MD and funnel meta dynamics nicely confirms the role of a key histidine.

The second part tries to address the fundamental question of what makes a NAM a NAM. To get at this, the authors use a series of simulations, but its not very clear exactly what was actually run - they say "ran molecular dynamics simulations of the DMCM complex with all ligands bound, or with DMCM removed from some or all its ECD or TMD sites" It looks from the figures like nothing bound, DMCM bound everywhere, DMCM just at the TMD and DMCM just at the END, but I am not entirely sure? Also - how many sims were performed for each etc?

We apologize for the lack of clarity in our statements about which sites were occupied by which ligands in which simulations. We have revised the text to state:

"We ran molecular dynamics simulations of the DMCM complex with all ligands bound, or with DMCM removed from either the ECD or TMD sites or both."

We state in the Methods section:

"After energy minimization and equilibration, three replicates of each system were simulated for 1 μ s using Gromacs 2019 and a time step of 2 fs."

The authors then say that the ECD RMSD was still very low but that there was a pattern of tertiary blooming consistent with closure of this family of receptors. This part of the work is very interesting, but the reader is still left wondering what it is about DMCM that makes the structure "bloom" and it feels like the authors kind of ran out of ideas at this point. One might think that NAMs are slightly bigger than PAMs for example - is this true? (I don't know) - or do they make a specific interaction? - In other words, what is that causes the change in the relative positioning of the ECD with respect to the TMD? I feel like this is super-important because really what we want is to have rules so we are able to say - "if you want a PAM, don't have X or Y etc". It feels like the authors are tantalizingly close

here, and it at the very least warrants a bit more in-depth discussion even if they cannot resolve this.

We agree with the reviewer that this is the frustratingly missing conceptual piece in the study. We show correlations between a ligand activity and conformational state but we lack an intuitive causative mechanism that could be generalized. Discovering and proving this mechanism would resolve many outstanding questions with diverse therapeutic applications. We have two ideas, one based on receptor determinants, and one based on ligand chemistry. An example of an important of receptor determinant is T142, which forms an H-bond with DMCM. Mutagenesis of this residue can convert NAMs and silent allosteric modulators (like flumazenil) into PAMs (PMID 7806498). The results are quite complex, however. Some PAMs become more efficacious and some less. These results suggest that rather modest changes in the ECD site can have rather large effects on efficacy. Interpretation becomes more problematic in the context of now understanding that many of these PAMs and NAMs bind both in the ECD and TMD, and can have opposing activities through the two classes of sites. Accordingly, it is likely impossible to define a general mechanism that distinguishes PAMs vs. NAMs based on specific ECD atomic interactions. Another more conceptual difference between PAMs and NAMs is that NAMs tend to be more rigid than PAMs. For example, flumazenil and the beta-carbolines share fused 3-ring structures that would be more effective as inflexible wedges at the subunit interface that through binding would prevent contraction of the ECD needed for channel activation. Bicuculline, the best known antagonist, shares this feature of a comparatively inflexible fused 3-ring structure (actually two fused 3-ring components). A more flexible structure would be more capable of first binding and then adopt a high affinity conformation associated with a compact, activated-state ECD conformation. There are holes in this concept, however. There are several benzodiazepines that are PAMs but comprise a fused 3-ring structure, for example bentazepam, bretazenil, and alprazolam. By comparing the binding modes of diazepam and alprazolam with DMCM, flumazenil and bicuculline, we found the benzodiazepine PAMs lack the extensive contacts with the complementary subunit seen with flumazenil and DMCM (and bicuculline at the β - α site). These “antagonists” are positioned in a way that could act like a doorstep to prevent the contraction of ECD, illustrated best in Supplementary Figure 5. We now include this idea in the main manuscript text.

Minor points

Fig 2. It says H-bonds are indicated with a dashed line. In my version of the pdf as least, I could only (just) see a dashed line in panel b between T207 and one of the imidazopyridine nitrogens (with what looks like poor h-bond geometry but could just be the view). Would be useful to make the h-bonds more obvious. It might be better to slightly move the view because the S205 h-bond talked about in the text is also not visible. And according to Supplemental 2e, these are the only two hydrogen bonds present.

We have modified Figure 2d (was 2b) accordingly.

Fig. 3. What are the coordinates on the X-Y axis?

These axes are in nanometers (nm); we now label them. Thank you for pointing out this omission.

Reviewer #3 (Remarks to the Author):

Zhu et al report cryo- EM structures of zolpidem- and DMCM- bound $\alpha 1\beta 2\gamma 2$ GABA-A receptor

constructs. The collaborators from Sweden contributed MD simulations of various ligand bound states, some of which are not accessible to the experiment: namely receptors with only some of the observed binding sites occupied. With the total body of data, a mechanism for negative and positive allosteric modulation by DMCM is put together from the static structures and the simulations. The subtype selectivity of zolpidem is examined in the light of the observed binding mode. In total, this is a very valuable study which adds to the knowledge of the mechanisms governing allosteric modulation. The results thus can be expected to pave the way towards a more rational development of compounds with desired profiles, as well as to future improved drugs.

There are some points of concern that should be addressed prior to publication as listed point by point:

The major concerns:

The functional data that was obtained with the used constructs and ligands is only presented in the supplementary figures, and mentioned “in passing”. It does, however, add value to the study, and raises questions that deserve to be addressed. The overall balance between the presentation of results obtained from the structures, the MD simulations and electrophysiology should be reconsidered, giving adequate room to the functional data in the MS and potentially compacting long, rather descriptive passages that deal with aspects of the presented structures.

We agree with this reviewer and reviewer 1, who commented on the same topic. We include new functional analysis now, and present it in main figures 2 and 4.

The questions raised by the functional data in detail:

Concern 1: Supplementary Fig 2a and accompanying MS text:

In the methods it is stated that experiments were done with $n \geq 3$. Where is the data beyond a single representative trace? The figure legend for 2a does not give the precise n for the representative observations. Mean values for the reference currents and the modulation are not reported. All data resulting from experiments described in the methods should be integrated. Line 100 (where the Suppl. figure 2a is called) states “native-like response” – this suggests a comparison with a more native set of constructs. Plz reword to reflect that the data corresponds with published observations, if this is what is meant. In panel a, the 4th trace shows the observed current in presence of 5 microMol GABA, 1 microMol Zolp and 3 microM FLM. This is interpreted as competitive inhibition on Zolp modulation by FLM – this should be demonstrated by a quantitative comparison between traces 2 and 4, using all datasets. Moreover: The samples for structure determination were prepared in presence of 10 microMol Zolp, and the functional effect of the presence of Zolp at the TMD sites is described as “remains unclear” in line 134. Perhaps an experiment similar to the one shown in the fourth trace but with higher Zolp concentration would help here to either rule out, or demonstrate that zolpidem modulates at higher concentrations in the presence of FLM. In Supplementary Figure 4a, the corresponding experiment with DMCM was performed at 10 microM DMCM.

We strongly agree with more emphasis being merited in the functional analysis. We now include analyses from recordings in independent cells in main Figures 2a, b and 4 a-d. We expand on our answer here in reply to Reviewer 1 on the same topic. We have clarified in the text as suggested regarding what we meant by “native-like” responses. We like the concept of using a high concentration of zolpidem and a high(er) concentration of FLM, however it will be challenging to interpret the results. FLM, on its own, at a high concentration, can cause potentiation as well (PMID: 2448430).

Concern 2: Supplementary Figure 4a and accompanying MS text:

The same issues as discussed for zolpidem apply to this dataset as well: Panels a and b provide representative traces, but the quantitative data analysis is missing. The legend states “representative results from n=3 recordings from different cells” – is a single representative trace per experiment shown? (Or averaged traces, which seems the less plausible interpretation of the legend text?) Please provide a summary of analyzed data, with all relevant comparisons (reference currents compared to the co-application experiments). Moreover: Panel b, second trace: It looks like the application was stopped during the rising phase, without reaching peak current. Another issue is the statement in line 226 “as well as directly activate the receptor (Supplementary Fig 4b)”: No experiment in the absence of GABA is displayed in Supplementary Fig 4b, please provide this data, or rewrite accordingly.

Thank you for these helpful suggestions and questions. As for zolpidem, we now have included the analysis of our DMCM-modulation recordings in main Figure 4. Regarding the comment about the ligand application not being long enough to reach the peak current, we tested longer applications of ligands, now shown in Figure 4c. These experiments give more confidence that peak currents measured previously were a good estimation. We observed apparent changes in desensitization kinetics as a function of DMCM concentration, which are reproducible. Specifically, 1 μ M DMCM results in faster desensitization, consistent with its binding to the high affinity ECD site having a NAM activity. With increasing concentrations (10, 100 μ M), desensitization slows or disappears, consistent with a superimposed PAM activity. While we think these differences are interesting, we did not do the more extensive experiments needed to rigorously quantify changes in kinetics, and as such do not comment in the manuscript on this aspect of the ligand activity.

Also, we have revised our manuscript and removed the statement “as well as directly activate the receptor”.

Related to the issues with the presentation of functional data for DMCM: Line 226 indicates “at low concentrations acts as an inverse agonist, or negative allosteric modulator” – presumably the intention is to state that it acts as negative allosteric modulator (also historically termed benzodiazepine receptor inverse agonist). Please rewrite using clear and IUPHAR conforming terminology. See also related minor point 7 below.

We have made this correction.

Concern 3: The subtype selectivity of zolpidem is explained by the observed binding mode and the absence or presence of specific ligand- protein interaction. This analysis would greatly benefit from a thorough comparison with the available structure of the α 5 subunit to shed light on the impact the loop B has on the ability of the ligand to bind with high affinity at α 1, α 2 and α 3, but not at interfaces formed by the so-called zolpidem insensitive subunits α 4, α 5 and α 6 which have a proline in the position homologous with α 1-T163 (discussed in lines 174ff). The “more complex” influence that the proline may have can possibly be deduced from the α 5 subunit’s structure – at least this should be attempted.

We presume the reviewer is referring to the cryo-EM structure of the receptor comprising one α 5 subunit and four β 3 subunits (PDB ID: 6A96), with nanobodies bound (and not the chimeric α 5 TMD/ β 3 ECD with neurosteroid bound from the Aricescu lab). While there are caveats related to different pentamer subunit compositions, most notably the lack of a γ subunit, we superimposed the α 5 subunit

(gray) with our zolpidem-bound $\alpha 1$ subunit (green). This superposition, in the region of the zolpidem site, is shown in the figure below. The relevant proline and threonine residues are shown as sticks, and zolpidem is positioned as modeled in the $\alpha 1\beta 2\gamma 2$ structure. The two residues are aligned well and their distance to the nearest atoms of zolpidem is ~ 7 Å. At this distance there is unlikely to be a direct effect from the side chain on the ligand sensitivity. Proline's side chain has the distinctive cyclic structure that gives proline an exceptional conformational rigidity compared to other amino acids. A speculative hypothesis is that proline's rigidity may allosterically affect the surrounding loops, such as loop C, and thereby disrupt the ligand binding, since zolpidem is exceptionally dependent on the local chemical environment for its high affinity binding.

Also concerning the determinants of zolpidem selectivity in loop C, the impact of $\alpha 1$ -V203 (lines 188-190) seems overstated, the EC₅₀ values in the zolpidem sensitive $\alpha 2$ and $\alpha 3$ – containing receptors are only slightly to the right of those for the $\alpha 1$ - case (see e.g. • DOI: 10.1016/j.ejphar.2010.03.015)

Thank you for drawing our attention to this study. We have revised this section in the manuscript to decrease the emphasis of $\alpha 1$ -V203 in determining zolpidem selectivity.

Minor points:

1: Abstract, line 26: Based on currently approved medications and their ATC codes, it seems inappropriate to list depression side by side with insomnia and epilepsy – no GABA-A R targeting medication bears an ATC code as antidepressant, but many as anxiolytics or hypnotics, and several as antiepileptics. Use in patients with depression is strictly off label at the time, and preclinical research into this direction has yet to provide compounds suitable for clinical trials as monotherapy in depression. (This point might seem a bit nit-picking, but it should be easy to provide a correct and balanced view of what is actually really approved and for which indication)

We greatly appreciate this detailed feedback and have removed the language about antidepressants.

2: Line 31: sedative should read sedative- hypnotic, or just hypnotic. The main indication of zolpidem is as sleeping aid, not as sedative.

Thank you for these suggestions. We now use “hypnotic” instead of “sedative” in the text.

3: Lines 45 and 52: “anxiety” is not a disorder. The correct term is “anxiety disorders”, while anxiety is a perfectly normal emotional response to threatening stimuli. (Again, this might seem overly picky – but the clinical scientists are, rightfully, irritated by such incorrect use of terminology that sadly is widespread in the preclinical literature.)

We now refer to “anxiety disorders.”

4: Lines 68-70: The end of the sentence should read “to enhance GABA- elicited currents or to elicit currents” as it’s stated that DMCM can act as allosteric agonist.

We have revised the text to: “to enhance GABA-elicited currents.”

5: Lines 145-148: Reference 33 reflects work in rodents. The notion that $\alpha 1$ -containing receptors mediate sedation in humans has been rejected (see doi: 10.1016/j.tips.2012.08.003.) – and still is often re-stated as a never verified myth in countless research articles and reviews.

This is indeed an important point. We have adjusted the text to highlight the discrepancy between rodent and human studies.

6: Line 215: Should this read “helping to form”?

Yes, thank you; corrected.

7: Lines 226, 249, 279: In most places in the article, the authors use the terms positive/ negative allosteric modulators, but in these places (and possibly elsewhere) they use the outdated terms “inverse agonist/ agonist” which are misleading as they can be confused with orthosteric ligand effects (historically, the full term was “inverse agonist/ agonist of the benzodiazepine receptor”). IUPHAR recommendation should be followed throughout, it can be stated in the introduction that benzodiazepine site PAM/ NAM compounds are often termed “inverse agonist/ agonist of the benzodiazepine receptor” – chiefly in older papers.

We agree that the inverse agonist terminology is confusing and have replaced it with PAM and NAM terminology.

8: Line 361: “ligands” is unspecific

We have added the relevant ligand concentration details.

9: Figure 5 and legend: The chain labels in panel b are hard to see, read and interpret. Perhaps “chain A” and “chain C” respectively would be better. Panel c: It is a bit puzzling to see that simulations with no ligand (None, orange plots) are quite different for the two panels with a biphasic appearance in the DMCM image, and a monophasic one in the right gallery. Some words of explanation might be helpful.

In terms of Fig 5b, we have revised the chain labels accordingly.

For panel c: The pLGICs are expected to exist in a dynamic ensemble of states with varying populations of twisted/untwisted configurations. With the removal of DMCM from both the ECD and TMD binding

sites, we believe the biphasic behavior to stem from the $\beta 2$ subunit dynamically interconverting between twisted and untwisted configurations characteristic of resting and activated configurations respectively.

However, we cannot discount the differential population in the DMCM simulations stemming from a subpopulation of $\beta 2$ subunits retaining memory of their initial configuration. Thus, an extended simulation (longer than the 1 μ s performed here) might converge the two systems without allosteric modulators into a similar distribution.

10: Figure legend for Fig 6, line 691: alpha-beta-gamma should either all be Greek letters, or all spelled out.

Thank you; we corrected the legend.

11: Figure 6: The cartoon aimed to depict the motions that lead to positive or negative allosteric effects is not particularly illustrative or intuitive, the “expanded” versus “contracted” ECD should be intuitively obvious. Several improvements of this cartoon can be suggested: (a) The legend should clearly explain why only 2 subunits are shown for the TMD, while three depict the ECD. (b) D-low and D-high need to be defined. The left image suggests that zolpidem binds with the same (high) affinity at ECD or TMD sites – or should it state Z-high? (c) Expanded and contracted ECDs should look differently.

Thanks for all these suggestions. We modified this figure and revised the legends based on your suggestions.

12: Supplementary Fig 2f: Amino acid labels to match panel e would be helpful.

Supplementary Fig 2e (previously 2f) is to show the map quality. Adding residue labels might interfere with viewing the EM map density. Main Figures 2d and 2e are the figures that show all the chemical interactions with the residue labels as well.

13: Same as remark 12, for panel d and c of Supplementary Figure 4.

The same as remark 12. Main Fig 4f and 4g have all the residues labeled.

14: Supplementary Figure 5, amino acid labels in panels a-d would be helpful.

The residues are labeled now in Supplementary Figure 5.

REVIEWERS' COMMENTS

Reviewer #1 (Remarks to the Author):

the author did a good job and addressed all my concerns.

Reviewer #2 (Remarks to the Author):

The reviewer's rebuttal and revised manuscript addresses my comments. I understand the need for brevity regarding the complexity of disentangling PAM/NAM determinants, but it would be good to say this explicitly - the authors have a paragraph that tries to expand on factors that may make NAMS, but I think its worth expanding that to state that deciphering the rules of PAM versus NAMS will be/is non-trivial.

Reviewer #3 (Remarks to the Author):

The authors have addressed all concerns, at this time, I recommend publication.

Congrats, margot ernst

Response to reviewers' comments:

Reviewer #1 (Remarks to the Author):

the author did a good job and addressed all my concerns.

Response: We thank this reviewer for their positive review of the revised manuscript and for their earlier constructive comments/requests.

Reviewer #2 (Remarks to the Author):

The reviewer's rebuttal and revised manuscript addresses my comments. I understand the need for brevity regarding the complexity of disentangling PAM/NAM determinants, but it would be good to say this explicitly - the authors have a paragraph that tries to expand on factors that may make NAMS, but I think its worth expanding that to state that deciphering the rules of PAM versus NAMS will be/is non-trivial.

Response: We agree and we appreciate this reviewer's suggestion (and their earlier critique). We have added the following statement to the main text:

"Comparison of the binding mode of DMCM with other benzodiazepine site modulators as well as the antagonist bicuculline suggest a bulky and rigid chemical core of the ligand, and extensive packing against the complementary subunit, may act in concert to interfere with contraction of the ECD necessary for channel activation, and thereby define important components of a negative modulator (Supplementary Fig. 5a-f). Further elucidation of general rules that determine PAM vs. NAM activity is of great interest and will be challenging."

Reviewer #3 (Remarks to the Author):

The authors have addressed all concerns, at this time, I recommend publication.
Congrats, margot Ernst

We thank Dr. Ernst for her helpful constructive critique of the first submission and the positive review of the revised manuscript.